# TCGEx: a powerful visual interface for exploring and analyzing cancer gene expression data

M Emre Kus [1], Cagatay Sahin [1,6], Emre Kilic[1,6], Arda Askin [1,6], M Mert Ozgur[2,6], Gokhan Karahanogullari[3], Ahmet Aksit[4], Ryan M O'Connell[5] & H Atakan Ekiz [1 ✉]

## Abstract

**Analyzing gene expression data from the Cancer Genome Atlas (TCGA) and similar repositories often requires advanced coding skills, creating a barrier for many researchers. To address this challenge, we developed The Cancer Genome Explorer (TCGEx), a user-friendly, web-based platform for conducting sophisticated analyses such as survival modeling, gene set enrichment analysis, unsupervised clustering, and linear regression-based machine learning. TCGEx provides access to preprocessed TCGA data and immune checkpoint inhibition studies while allowing integration of user-uploaded data sets. Using TCGEx, we explore molecular subsets of human melanoma and identify microRNAs associated with intratumoral immunity. These findings are validated with independent clinical trial data on immune checkpoint inhibitors for melanoma and other cancers. In addition, we identify cytokine genes that can be used to predict treatment responses to various immune checkpoint inhibitors prior to treatment. Built on the R/Shiny framework, TCGEx offers customizable features to adapt analyses for diverse research contexts and generate publication-ready visualizations. TCGEx is freely available at https://tcgex.iyte.edu.tr, providing an accessible tool to extract insights from cancer transcriptomics data.**

**Keywords** TCGA; Functional Genomics; Machine Learning; Interactive Analysis; Gene Expression
**Subject Categories** Chromatin, Transcription & Genomics; Computational Biology; Methods & Resources

## Introduction

The Cancer Genome Atlas (TCGA) launched in 2006 is among the most ambitious projects to better understand and tackle human cancer (Blum et al, 2018). As of this writing, more than 11,000 tumor samples and matched healthy tissues have been

characterized at the molecular level across 33 different cancer types as part of this initiative. TCGA studies employ a variety of experimental approaches including RNA and microRNA (miRNA) sequencing, whole-exome sequencing, and genotyping and methylation arrays to generate 2.5+ petabytes of publicly available data. Landmark studies from the TCGA network described common genetic aberrations in cancer such as TP53 loss (ICGC/TCGA Pan-Cancer Analysis of Whole Genomes Consortium, 2020) as well as other cancer-specific changes including BRCA1/2 mutations in breast and ovarian tumors (Bell et al, 2011; Koboldt et al, 2012) and APC mutations in colorectal carcinoma (Muzny et al, 2012). Furthermore, transcriptomic profiling has led to the characterization of immune signatures within the tumor microenvironment (TME). Follow-up studies have revealed that the genomic and transcriptomic features of tumors determine whether they will respond to targeted therapies and immunotherapeutics (Van Allen et al, 2015; Hugo et al, 2015; Ayers et al, 2017; Auslander et al, 2018), suggesting that a better understanding of the heterogeneity within the TME is essential for improved therapeutics and clinical outcomes.

Analyzing genomic data in the TCGA and other public repositories effectively requires programming skills and it is often challenging for the non-specialists. Multiple web-based applications have been developed to facilitate TCGA data analysis, including cBioPortal, Web-TCGA, UCSC Xena, and GEPIA2 (Cerami et al, 2012; Deng et al, 2016; Goldman et al, 2020; Tang et al, 2019). Although these visual analysis interfaces reduced the barriers to data utilization for many investigators in the field, there is still a need for a centralized platform that integrates multiple methods for customized analyses. In particular, no interactive tool to date is equipped with sophisticated functional genomic analysis capabilities that allow examining transcriptomics data from various sources including TCGA, landmark immunotherapy studies, and one's own research. To address this need and democratize the use of high-throughput data in the cancer research field, we introduce The Cancer Genome Explorer (TCGEx), an open-source web-based analysis platform written in R/Shiny that offers several modules for sophisticated custom analyses. Using this novel interface, we investigated the TCGA skin cutaneous melanoma (SKCM) data and identified gene expression patterns and mutational subtypes

[1]The Department of Molecular Biology and Genetics, Izmir Institute of Technology, 35430 Gulbahce, Izmir, Turkey. [2]The Department of Molecular Biology and Genetics, Bilkent University, 06800 Cankaya, Ankara, Turkey. [3]The Department of Mathematics, Izmir Institute of Technology, 35430 Gulbahce, Izmir, Turkey. [4]The Department of Information Technologies, Izmir Institute of Technology, 35430 Gulbahce, Izmir, Turkey. [5]The Department of Pathology, University of Utah, Salt Lake City, UT 84112, USA. [6]These authors contributed equally: Cagatay Sahin, Emre Kilic, Arda Askin, M Mert Ozgur. ✉E-mail: atakanekiz@iyte.edu.tr

associated with differential survival. We also examined intratumoral immune signatures using TCGEx machine learning modules and revealed miRNAs associated with elevated interferon-gamma (IFNγ) signaling. We validated these findings on independent data sets from immune checkpoint inhibition (ICI) trials and further identified a cytokine signature associated with positive responses to ICI across multiple types of cancers.

TCGEx features ten analysis modules that can be flexibly adapted to various study contexts. Integrative analyses by aggregating data from multiple projects are also possible to investigate commonalities and differences among cancers. TCGEx's comprehensive computational pipeline utilizes preprocessed data from the TCGA repository that was prepared by combining transcriptomics data with clinical metadata and intratumoral immune signatures described by the leading studies in the field (Thorsson et al, 2018). Additionally, TCGEx features transcriptomic data from multiple landmark ICI studies (Van Allen et al, 2015; Gide et al, 2019; Riaz et al, 2017; Hugo et al, 2016; Zhao et al, 2019; Cloughesy et al, 2019; Miao et al, 2018; Liu et al, 2019; Kim et al, 2018; Rosenberg et al, 2016; Balar et al, 2017; McDermott et al, 2018; Choueiri et al, 2016) and it allows users to upload and analyze their own data sets. We provide step-by-step in-app tutorials to increase user-friendliness and help researchers implement TCGEx into their workflows and generate publication-ready plots. TCGEx is publicly available without requiring user registration at https://tcgex.iyte.edu.tr web server as well as through a docker image to facilitate local execution and open-source program development (https://hub.docker.com/r/atakanekiz/tcgex). The source code of TCGEx is accessible at https://github.com/atakanekiz/TCGEx.

## Results

TCGEx interface enables exploring the gene expression landscape in cancer and provides access to numerous functional genomics analysis pipelines. Its modular code structure allows interaction with preprocessed TCGA data as well as data from other sources (Fig. 1). Distinctive features of TCGEx complement the available tools in the field and provide access to new analysis capabilities (Table EV1). In this study, we have used this tool to investigate the transcriptomic landscape of human melanoma and other cancers with the goal of identifying gene expression patterns associated with antitumor immunity and immunotherapy response. To demonstrate the platform's capabilities and show how it can be adapted by users with different research hypotheses, we focus on two different classes of genes in particular, miRNAs and cytokine/cytokine receptor genes. All the figures herein are obtained from the TCGEx interface unless noted otherwise. Melanoma is an aggressive form of cancer that develops in pigment-producing cells of the skin, although it can rarely be observed in other organs including the eye and gastrointestinal tract as well. More than 320,000 new cutaneous melanoma cases are reported globally each year, and 55,000+ lives are lost because of this disease (Sung et al, 2021). UV exposure is a significant risk factor for the development of melanoma, and most cases are diagnosed at an advanced stage where the cancer cells have already metastasized to nearby lymph nodes or distant organs. Accordingly, the TCGA-SKCM study is composed of mainly metastatic tumor biopsies (368 metastatic, 103 primary tumors), as shown in the TCGEx data selection module

(Fig. 2A). These samples originated from comparable numbers of male and female patients in various age groups (Fig. 2B). The landmark TCGA paper describing the genetic features of SKCM classified tumors into subtypes based on mutations in BRAF, NF1, and RAS genes (Cancer Genome Atlas Network, 2015). These subtypes are characterized by distinct sets of accompanying genomic aberrations including PTEN loss in BRAF-mutant tumors, KIT amplification in triple wild-type (WT) tumors, Akt3 amplification in RAS-mutant and wild-type tumors, and upregulation in ITGB8 in NF1-mutant tumors which are readily observed in transcriptomics data (Fig. 2C). The expression profiles of these genes were largely consistent between metastatic and primary tumors; however, the smaller sample size in the primary tumor group resulted in a greater variability in the data (Fig. EV1A). When these graphs are faceted by patient sex, some of these genes showed sex-dependent differences in their expression patterns (Fig. EV1B,C). Interestingly, BRAF-mutant melanomas expressed higher levels of transferrin (TF) which was recently shown to protect circulating tumor cells from oxidative stress leading to BRAF inhibitor resistance (Fig. EV1D) (Hong et al, 2021). On the other hand, NF1-mutant melanoma was distinctly marked by the high expression of hepatocyte-specific transcription factor ONECUT1 which was shown to be associated with tumor progression (Fig. EV1D) (Zhao et al, 2014). Our analyses also revealed that ZC3H13, a component of the RNA methyltransferase complex, is enriched in RAS-mutant melanoma; and triple-WT tumors selectively expressed FOXF2, a transcription factor that plays a complex role in regulating oncogenic signaling pathways (Fig. EV1D) (He et al, 2020). These findings suggest that distinct molecular mechanisms may be responsible for driving different tumor subtypes in melanoma and indicate possible intervention points.

BRAF-mutant melanoma constitutes about 50% of the cases in the TCGA cohort followed by RAS-mutant (~30%), NF1-mutant, and triple-WT tumors (~10% each). Of these four groups, BRAF-mutant melanoma has the most favorable prognosis (Fig. 2D), although this may be confounded by the younger age of the patients in this group (Cancer Genome Atlas Network, 2015). While melanoma incidence is comparable between both sexes, females were reported to have a reduced risk of mortality (Morgese et al, 2020). Examination of the TCGA-SKCM cohort as a whole did not show a significant survival difference between the two groups; however, a slightly increased risk was observed for the males (Appendix Fig. S1a,b). Notably, when we examined only BRAF and NRAS-mutant tumors, males were associated with poorer survival outcomes, especially in the NRAS-mutant subgroup (Fig. 2E). These analyses conducted on the entire SKCM cohort were repeated separately for metastatic and primary tumors, and possibly due to low sample numbers, clear survival differences were not observed between mutational subtypes in primary tumors (Appendix Fig. S1c–f). Taken together our findings suggest that sex-specific factors may differentially influence the prognosis in different mutation subtypes.

In addition to defining potential genetic drivers in melanoma, the TCGA-SKCM manuscript also identified three distinct tumor subsets based on gene expression profiles immune, keratin, and MITF-low groups (Cancer Genome Atlas Network, 2015). The immune subset was characterized by the higher expression of immune cell-specific genes (Fig. 3A), while the keratin and

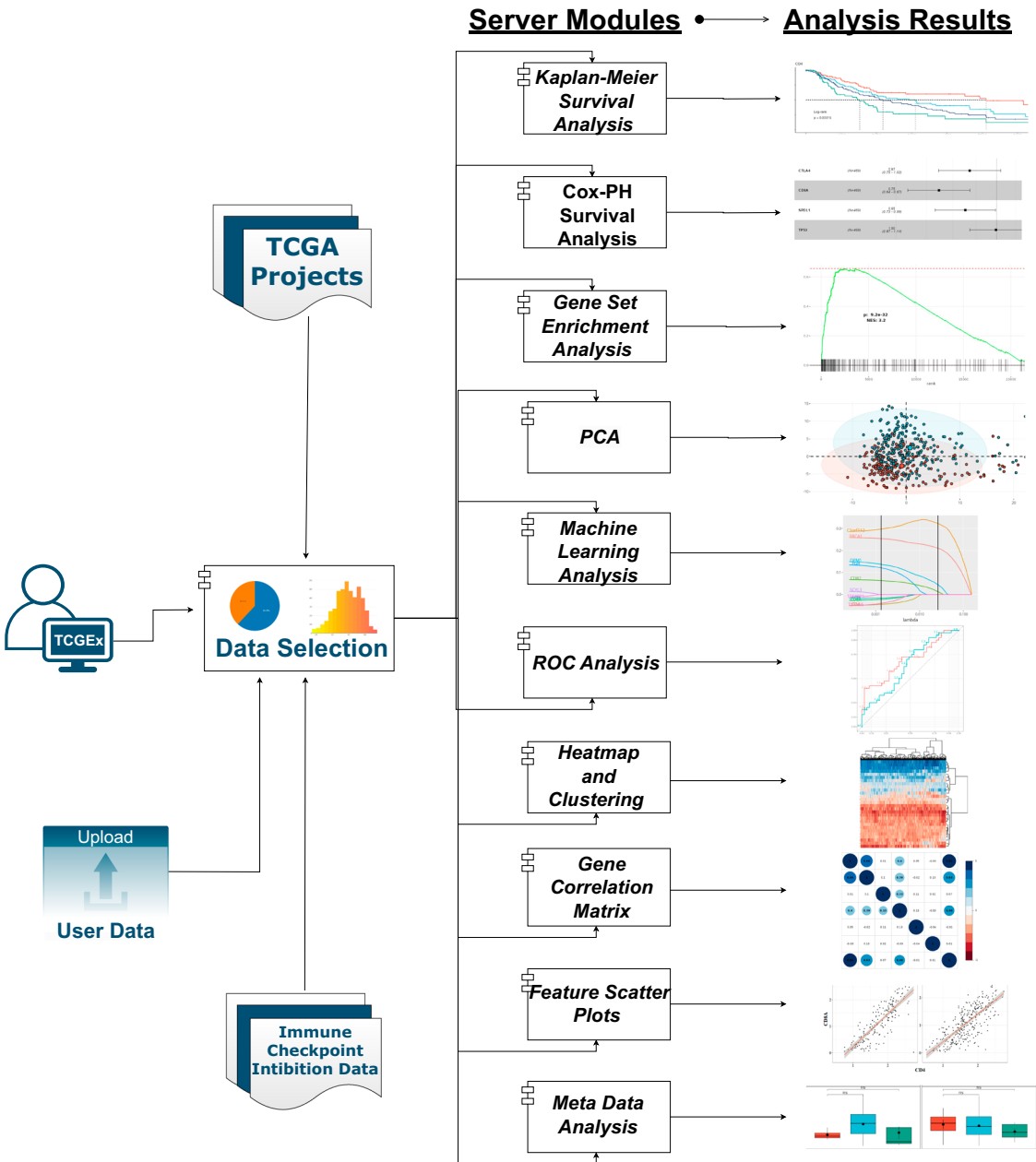

**Figure 1. TCGEx workflow overview.**

TCGEx pipeline begins with the user selection of preprocessed transcriptomics data from a single TCGA project or multiple projects for integrated analyses. The data selection module shows descriptive statistics including the sample types, sex, and age distribution of the patients in the selected project(s). The data are subsequently passed onto ten distinct modules for specialized analyses including survival modeling, gene set enrichment analysis (GSEA), principal component analysis (PCA), correlation calculation, and exploratory graphing in the form of scatter plots, boxplots, and heatmaps. In each module, the user can further focus on specific sample types such as primary and/or metastatic tumor samples, and some modules allow further subsetting in the analyses. Algorithmic and graphical parameters are easily changed through visual and responsive user interfaces. TCGEx modules are supported by interactive tutorials and provide options to export analysis results and graphical outputs for publication purposes. TCGEx is freely available on the web as well as through a docker image. All the subsequent figures in this manuscript are generated using the TCGEx modules.

MITF-low subsets were characterized by the high and low expression of epithelial genes, respectively, in the absence of a clear immune infiltration signature. Of these three groups, the immune-enriched subset exhibits the most favorable survival outcome as analyzed in the RNAseq and the RPPA data (Fig. 3B; Appendix Fig. S2a–d), which is consistent with the positive role of

inflammation in SKCM reported previously (Thorsson et al, 2018). IFNγ is a key inflammatory cytokine released from T and NK cells to eliminate tumor cells. IFNγ exerts its multifaceted functions within the TME by signaling through IFNγ-receptors (IFNGR) and regulating dozens of downstream genes in tumor cells and the neighboring immune cells (Parker et al, 2016). We examined IFNγ

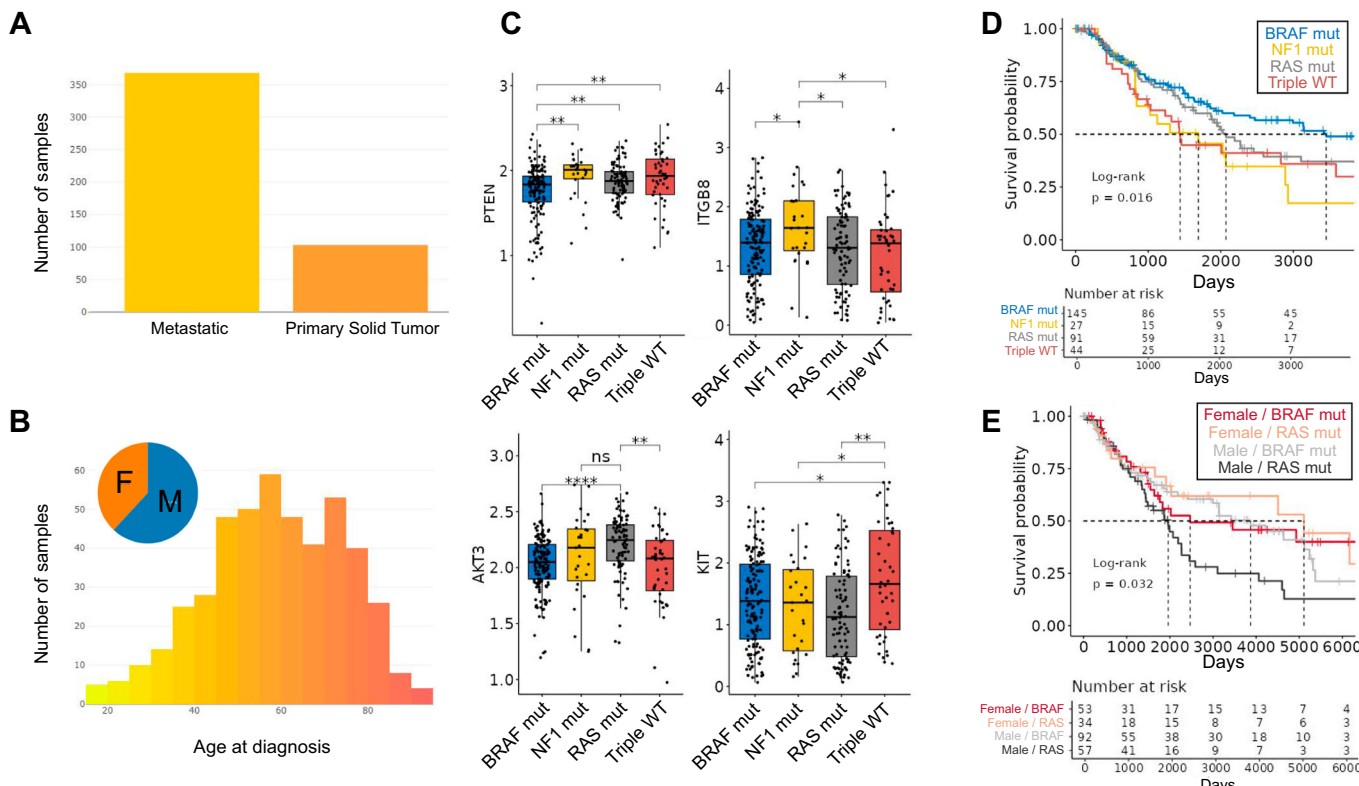

**Figure 2. Analysis of mutational subsets in skin cutaneous melanoma.**

(A) The TCGA-SKCM project consists of mostly metastatic samples. Analyses described herein are performed on both metastatic and primary tumors. (B) Comparable numbers of male and female individuals from a range of age groups are included in the SKCM patient cohort. (C) Expression profiles of PTEN, ITGB8, AKT3, and KIT can distinguish mutational subtypes of SKCM (BRAF mutants $n = 146$, NF1 mutants $n = 27$, RAS mutants $n = 91$, Triple WT $n = 44$), suggesting different molecular signaling landscapes exist in disease subsets (Cancer Genome Atlas Network, 2015). Statistical comparisons are performed using the $t$ test, although users have the option to select nonparametric Wilcoxon test here as well. The box represents the interquartile range (IQR), spanning from the 25th percentile (lower bound) to the 75th percentile (upper bound). The horizontal line within the box indicates the median (50th percentile). Whiskers extend to the smallest and largest values within 1.5 × IQR from the lower (Q1) and upper (Q3) quartiles, respectively. The P values are adjusted for multiple comparisons (****$P < 0.0001$; **$P < 0.01$, *$P < 0.05$, ns$P > 0.05$). (D) Kaplan–Meier survival analysis revealed that BRAF-mutant SKCM has the most favorable clinical outcome. Ten-year survival curves are shown where dashed lines indicate the median survival time points in each group. The risk table provided below the survival curve denotes the number of surviving patients (i.e., at-risk patients) at different follow-up times. The log-rank P value shown on the graph tests the null hypothesis that none of the groups are different from one another. (E) Survival modeling using only BRAF- and RAS-mutant melanoma patients suggests that males in the RAS-mutant group have poorer survival outcomes compared to BRAF-mutant males and females in general. TCGEx provides users with flexible options to change the color scheme and the range of time axis in the graphs.

response gene expression signature in SKCM subsets and observed higher levels of most of these transcripts in the immune subset (Fig. 3C) (Liberzon et al, 2011), and as expected, these trends were paralleled by the elevated IFNγ transcript levels and CD8 + T-cell signatures within the TME. We next wanted to investigate the relationship between the intratumoral CD8 + T-cell score and the T-cell receptor (TCR) diversity (Thorsson et al, 2018). This is of interest because multiple studies have suggested that the clonally expanded T cells within melanoma are associated with a spectrum of dysfunctional phenotypes (Tirosh et al, 2016; Li et al, 2019). Increasing CD8 + T-cell signature within the TCGA-SKCM tumors positively correlated with the overall TCR diversity, suggesting a polyclonal T-cell infiltrate and the presence of T-cell clones with different antigenic specificities (Fig. 3D) (van der Leun et al, 2020). Notably, the increased TCR diversity correlated with higher levels of cytotoxic effector molecule perforin (PRF1) and decreasing genetic signatures of M2 macrophages that are known to exert pro-tumorigenic functions (Fig. 3E) (Thorsson et al, 2018). Taken

together, TCGEx can be utilized to explore multifaceted data types in TCGA including the tumor-specific subtypes and the intratumoral immune landscape.

MicroRNAs (miRNAs), 20–22 nucleotide noncoding RNAs responsible for post-transcriptional modulation, are key regulators of antitumor immunity and tumor immunoevasion (Xing et al, 2021). Given the importance of IFNγ signaling in cancer, we next focused on TCGA-SKCM miRNAseq data and utilized lasso regression to select top miRNAs associated with the IFNγ response. The patient cohort was divided into training (70%) and test (30%) subsets and the expression of miRNAs (529 genes after low-expression filtering) were modeled as the predictor variables against the average expression of IFNγ signature genes (MSigDB Hallmark IFNγ response gene set) as the response variable. In all, 26 positive and 18 negative coefficients remained in the optimally regularized model (log lambda + 1 standard error) (Figs. 4A and EV2A,B). Five miRNAs, miRs-155-5p, -150-3p, -150-5p, -142-3p, and -7702 were the last five predictors remaining

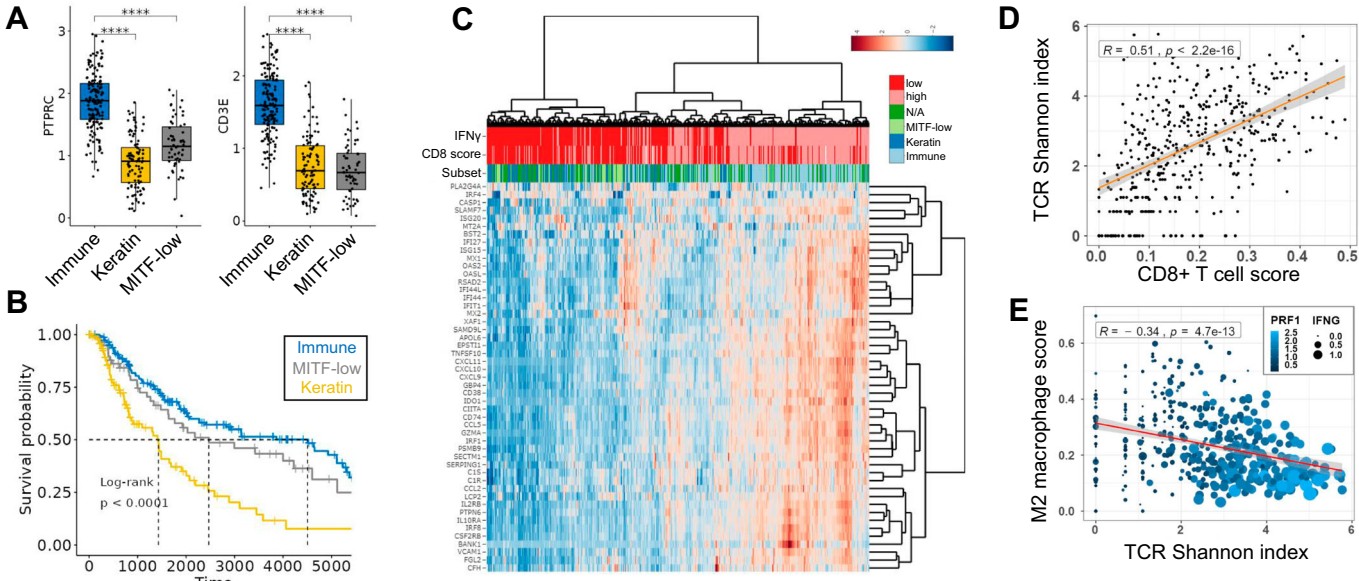

Figure 3. Analysis of the melanoma intratumoral immune landscape.

(A) The immune-enriched melanoma subset previously defined based on multi-gene expression profiles (Cancer Genome Atlas Network, 2015) is characterized by higher levels of PTPRC (CD45, a pan-leukocyte marker) and CD3E (a T-cell marker) transcripts (Immune $n = 163$, Keratin $n = 101$, MITF-low $n = 58$). The box represents the interquartile range (IQR), spanning from the 25th percentile (lower bound) to the 75th percentile (upper bound). The horizontal line within the box indicates the median (50th percentile). Whiskers extend to the smallest and largest values within $1.5 \times$ IQR from the lower (Q1) and upper (Q3) quartiles, respectively. T test P values are adjusted for multiple comparisons ($P < 0.0001$, "****"). (B) Immune subset of the SKCM is associated with an improved survival outcome compared to keratin and MITF-low subsets. (C) Heatmap visualization of the top 25% variably expressed genes from the MSigDB Hallmark IFNγ response gene set (Liberzon et al, 2011) in primary and metastatic melanoma is shown (50 out of 200 genes are plotted). The annotation rows on top of the heatmap were prepared by categorizing IFNγ transcript level and CIBERSORT CD8 + T-cell score (Newman et al, 2015) at their median values. Melanoma subsets defined previously based on genome-wide expression patterns are also shown. CPM-normalized and log-transformed transcript counts are scaled by row to better show expression differences in data subsets. The default TCGEx parameters were used for hierarchical clustering (Euclidean distance and Ward.D2 clustering algorithm for both genes and samples). (D) Increasing intratumoral CD8 + T-cell signature is associated with a higher TCR diversity suggesting a polyclonal T-cell infiltrate. The Pearson correlation test was used for statistical analysis. (E) The higher TCR diversity in melanoma tumors correlates with increased levels of T-cell effector genes including perforin (PRF1) and IFNγ and decreased pro-tumorigenic M2 macrophage signatures (Thorsson et al, 2018). Statistical analysis was performed using the Pearson correlation test, and the corresponding P values are shown.

in the model when the model penalty term lambda (λ) is increased further at the expense of minimized mean-squared error. Studies have suggested that these miRNAs can indirectly control the expression of immune effector molecules by acting on various mRNA targets (Hu et al, 2022; Trifari et al, 2013; Rodríguez-Galán et al, 2018). In line with this view and prior work, IFNγ, perforin, and granzyme B transcripts showed a positive correlation with these microRNAs in melanoma TME (Fig. 4B). In addition, known or predicted mRNA targets of miRNAs can be surveyed using TCGEx. However, it is important to note that a consistent negative correlation of mRNA and miRNA transcripts may not be always observable due to selective and/or incomplete targeting and obscuration of cellular-level interactions in bulk expression data. Of note, top positive correlators of miR-155-5p were immune-specific genes such as markers of T cells and costimulatory receptors; and the top negative correlators included regulators of nucleotide biosynthesis, metabolism, and signal transduction (Fig. EV2C–E). Since a single miRNA can control dozens of targets and miR-155-5p correlates with immunity in SKCM (Ekiz et al, 2019), we next examined whether the expression levels of miR-155-5p can distinguish the immune subset of SKCM from the keratin and MITF-low subsets. We performed ROC analysis and found that miR-155-5p predicted the immune-enriched SKCM

subset with an AUC of 0.826, although the average expression of IFNγ response genes was a slightly better predictor with the AUC of 0.887 (Fig. 4C). However, we noted that the predictive power of the averaged expression of 5 miRNAs that remained in the highly regularized lasso model was a better predictor of the immune subset of SKCM (AUC: 0.913) (Fig. 4D). Combining these miRNAs with the IFNγ response genes did not improve the prediction of the immune-enriched melanoma subset (AUC: 0.893) (Table EV3). To further examine gene expression patterns in miR-155-5p-high melanoma, we performed GSEA using annotated KEGG gene sets curated in the MSigDB database (Liberzon et al, 2011) and noted that the expression of genes associated with immunity was enriched in the miR-155-5p-high subset (Figs. 4E,F and EV2F). Lastly, we performed dimensionality reduction using 685 genes belonging to the "adaptive immune response" gene ontology class and color-coded the individual samples based on miR-155-5p expression to see whether miR-155 levels can differentially mark samples in the PCA space. Interestingly, the first principal component distinguished a portion of the samples suggesting that miR-155 is a marker of adaptive immunity signature within the TME which is supported by the work from our group and others (Fig. 4G) (Ekiz et al, 2019; Hsin et al, 2018; Gracias et al, 2013; Ji et al, 2015).

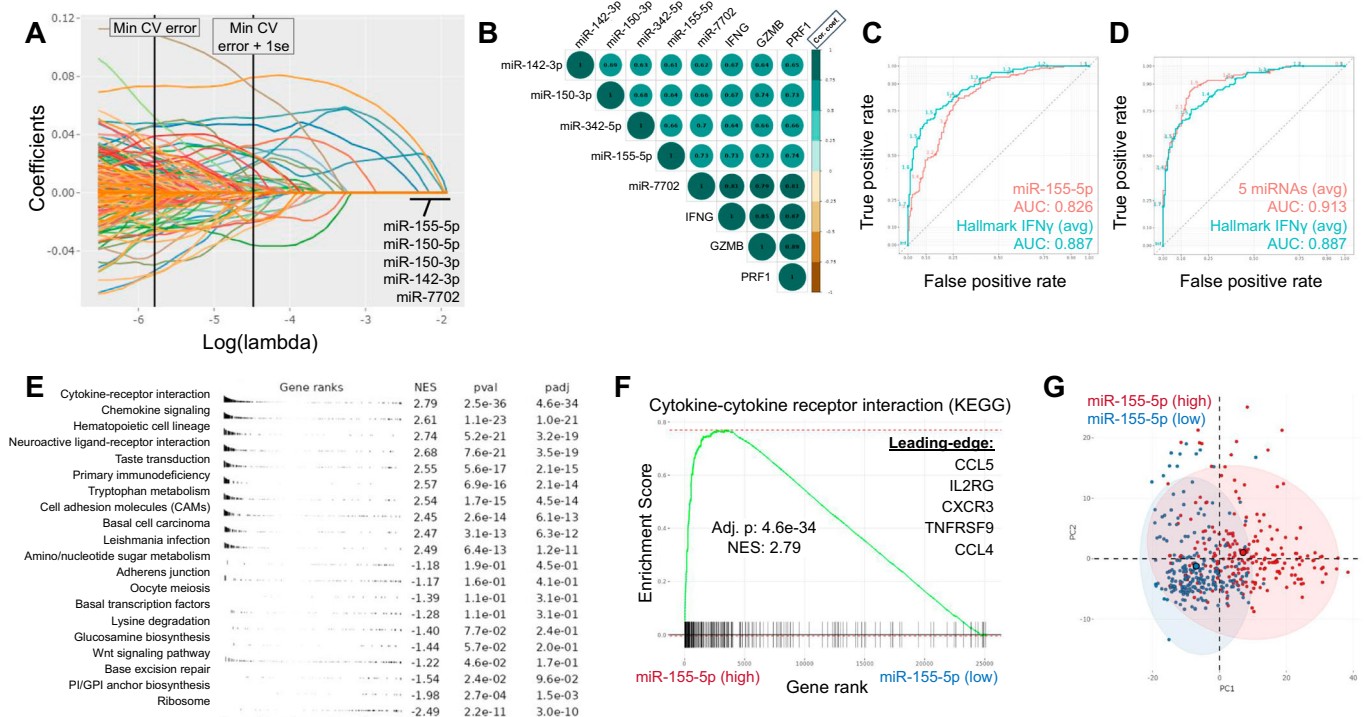

**Figure 4. Machine learning analysis of the associations between miRNAs and antitumor immunity in TCGA-SKCM data.**

(**A**) Lasso regression was performed by selecting IFNγ response score as the response variable and mature miRNA transcripts as predictor variables. In total, 529 miRNA predictors in the preprocessed SKCM data entered the lasso model after splitting the data set into training (70%) and test (30%) subsets for cross-validation. The graph shows the coefficient regularization paths for individual predictors at increasing levels of penalty term lambda (λ). Vertical lines indicate the cutoffs where the cross-validation (CV) error was at its minimum value and within one standard error. Five miRNAs that continued to have nonzero coefficients at arbitrarily high levels of regularization are shown. (**B**) Five miRNAs highlighted in (**A**) show positive correlations with each other and immune effector transcripts IFNγ, PRF1, and GZMB. Circle color gradient and size indicate pairwise Spearman correlation coefficients. (**C**) ROC analysis with miR-155-5p and averaged Hallmark IFNγ response gene set was performed to test their power in predicting the immune subset of SKCM (binarized as 1 for correct grouping) over keratin and MITF-low groups (both binarized as 0 for incorrect grouping). AUC values indicate both predictors are strongly associated with the immune subset, although the IFNγ response gene set performed better compared to miR-155-5p. (**D**) ROC analysis was performed as described in (**C**), with the averaged expression of five miRNAs (and IFNγ response) as predictors. In this analysis, the five miRNA expression signatures performed slightly better than the IFNγ response score for predicting the immune subset in melanoma. (**E**) GSEA was performed between miR-155-5p-high and -low subsets of melanoma (categorized at the median expression value) using genome-wide signal-to-noise ranked (Subramanian et al, 2005) transcripts and the KEGG gene sets (Liberzon et al, 2011). The top ten positively and negatively enriched gene sets are shown in the table where NES > 0 indicates enrichment in the miR-155-5p-high subset, and statistical significance was assessed using a permutation test, "padj" shows the Benjamini–Hochberg adjusted P value to account for multiple comparisons across the gene set. (**F**) The GSEA plot shows a significant enrichment of genes associated with cytokine signaling in the miR-155-5p-high melanoma samples. The leading-edge genes contributing strongly to the score are also shown. Statistical significance was assessed using a permutation test and, P values were adjusted for multiple comparisons using the Benjamini–Hochberg method. (**G**) Principal component analysis of melanoma samples was performed using 685 genes associated with the adaptive immune response GO annotation. Samples are categorized at the median expression of miR-155-5p and color-coded to demonstrate miR-155-5p-high and -low samples in the PCA space.

Next, we wanted to validate the associations between miRNAs and intratumoral IFNγ signature in independent data sets obtained from melanoma biopsies prior to ICI therapy (Van Allen et al, 2015; Gide et al, 2019; Riaz et al, 2017; Hugo et al, 2016; Liu et al, 2019). No miRNA sequencing has been done in these clinical trials yet, but the miRNA host gene expression is available in the RNAseq data. We observed that the expression of the mature miRNAs and their host genes show a strong positive correlation in the TCGA data (Appendix Fig. S3), and thus we focused on the miRNA host genes in the ICI data sets as a surrogate for mature miRNA expression. Using quartile-normalized integrated data from 5 different melanoma ICI studies (313 total samples prior to treatment with anti-CTLA4 or anti-PD1), we constructed lasso regression against the averaged IFNγ signature genes as the response variable and 45 miRNA genes that are expressed in at least 25% of the samples as

the predictor variables (Fig. 5A). This model predicted 11 miRNAs with positive coefficients, including MIR155HG, and 2 miRNAs with negative coefficients at the maximum penalization within 1 standard error of the minimum cross-validation error (Fig. 5B,C). We showed that MIR155HG expression is positively correlated with the cytotoxic lymphocyte markers CD8A, IFNG, and PRF1 across melanoma ICI data sets (Fig. 5D). As expected, the GSEA analysis revealed an enrichment of IFNγ signaling in MIR155HG-high samples (Fig. 5E). After validating MIR155HG's association with intratumoral immunity in melanoma ICI data, we performed similar analyses on pre-treatment Pan-ICI data prepared by integrating 10 quartile-normalized ICI studies (5 melanoma, 3 kidney, 1 stomach, and 1 bladder cancer studies adding up to 938 samples prior to treatment with anti-CTLA4, anti-PD1 or anti-PD-L1 inhibitors) (Van Allen et al, 2015; Gide et al, 2019; Riaz et al, 2017;

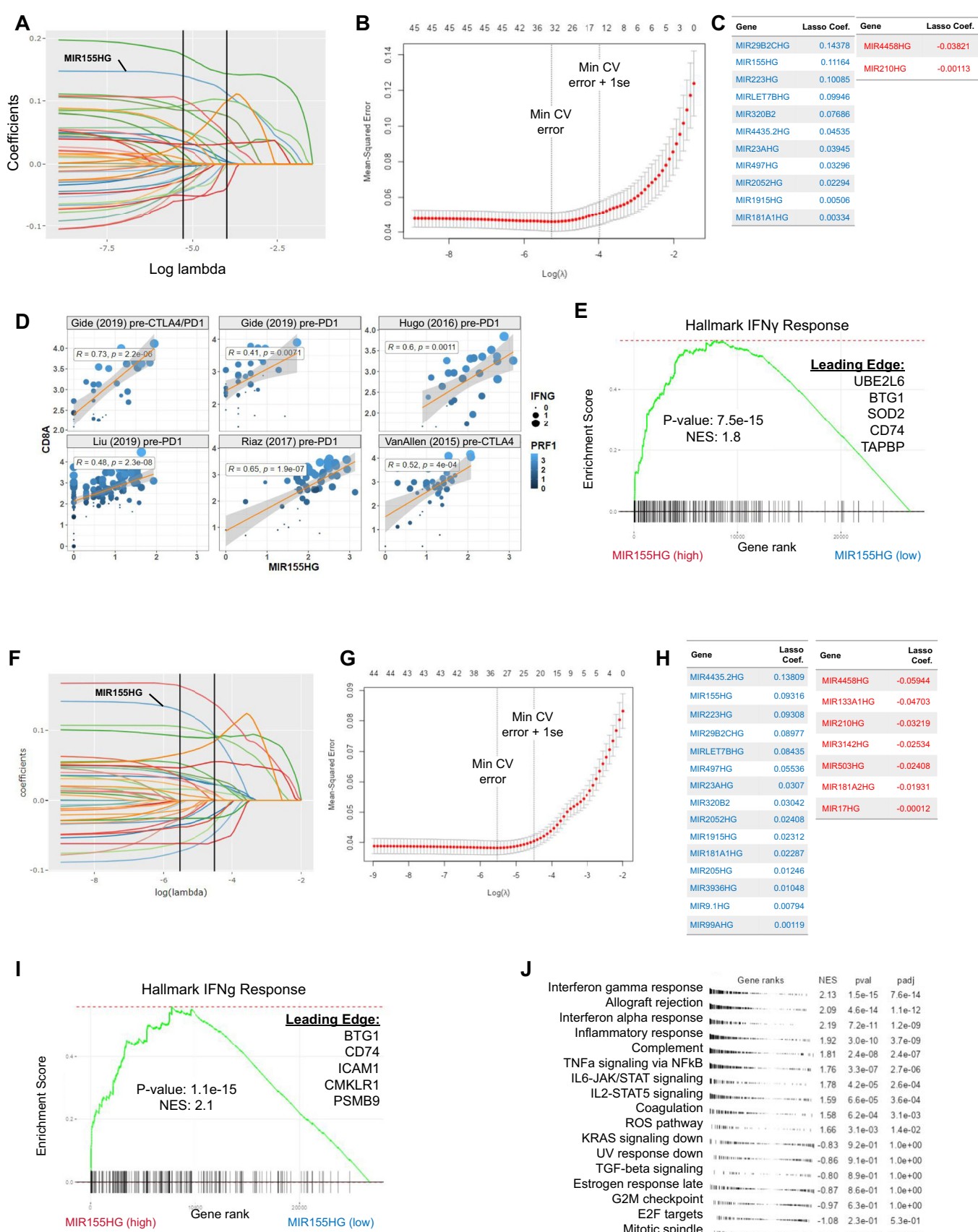

 **Figure 5. Investigating the associations between miRNAs and immunity markers in cancer immunotherapy studies.**

(A–C) Lasso regression was performed using the average expression of MSigDB Hallmark IFNg response gene set as the response variable and miRNA host genes as predictors in melanoma biopsies prior to immunotherapy ($n = 313$). The penalized coefficients (A) and the mean-squared cross-validation error of penalized models are shown (B). miRNA predictors with positive and negative coefficients at the lambda value corresponding to within 1 standard error of the minimum cross-validation error are listed (C). (D) Scatter plots show the relationship between MIR155HG and cytotoxic T lymphocyte-specific gene expression in melanoma ICI clinical trial data sets (Van Allen et al, 2015; Riaz et al, 2017; Hugo et al, 2016; Liu et al, 2019). Statistical analysis was performed using the Pearson correlation test, and the corresponding p values are shown. (E) Gene set enrichment analysis reveals higher expression of IFNγ-responsive genes in MIR155HG-high melanoma biopsies prior to ICI therapy. Statistical significance was assessed using a permutation test. (F–H) Lasso regression was performed using miRNA host genes as predictors and the average expression of MSigDB Hallmark IFNγ responsive genes as a response variable across pan-ICI data prepared by integrating ten different ICI studies comprising of 938 samples prior to treatment. Feature coefficients (F), mean-squared error of the penalized models (G), and the genes with nonzero coefficients (at minimum lambda + 1 standard error) are shown in immunotherapy biopsies ($n = 938$) (H). (I) MIR155HG-high biopsies demonstrate a significant enrichment of IFNγ-response gene signature in the pan-ICI data set. Statistical significance was assessed using a permutation test. (J) The top differentially enriched pathways are shown for the MIR155HG-high and MIR155HG-low biopsies prior to ICI treatment. Statistical significance was assessed using a permutation test and, P values were adjusted for multiple comparisons using the Benjamini–Hochberg method.

Hugo et al, 2016; Miao et al, 2018; Kim et al, 2018; Balar et al, 2017; McDermott et al, 2018; Choueiri et al, 2016; Liu et al, 2019). Lasso modeling against averaged IFNγ-responsive genes revealed 15 (including MIR155HG), and 7 miRNA genes with positive and negative coefficients, respectively (Fig. 5F–H) which were mostly consistent with the melanoma-specific analyses. When the genome-wide expression patterns of the pan-ICI pre-treatment data were examined, we noted that MIR155HG-high subset was associated with an enrichment of IFNγ-responsive gene set (Fig. 5I) as well as other pathways associated with the immune response (Fig. 5J). Taken together, machine learning-based analysis of melanoma and other tumor biopsies revealed the miRNAs associated with antitumor immunity and further supported the connection between intratumoral IFNγ signaling and MIR155HG. We provide step-by-step instructions to reproduce Figs. 4 and 5 to assist users in performing similar analyses (Appendix). As these data originate from bulk methods, it is not possible to ascertain the gene expression to specific cell types within the TME, however, there is literature supporting the immunoregulatory roles of MIR155HG in both tumor cells and the infiltrating immune cells (Curti et al, 2022; Hsin et al, 2018). To examine which cell types within the melanoma TME could express MIR155HG and other ML-identified miRNAs, we turned to publicly available single-cell RNA data hosted on the Broad Institute Single Cell Portal (Tarhan et al, 2023). We examined three data sets featuring immune cells and malignant cells from primary and metastatic melanoma biopsies (Tirosh et al, 2016; Jerby-Arnon et al, 2018; Teijeira et al, 2019). Not all miRNAs were found in these data sets possibly due to low-expression levels, and the detected miRNAs showed variable expression patterns across malignant and immune cell clusters (Fig. EV3A,C,D,G–I). While some miRNAs were broadly expressed (e.g., MIR17HG, MIR210HG, and MIRLET7BHG), MIR155HG expression was selective to the infiltrating T cells (Fig. EV3B,E,J). In particular, the highest MIR155HG expression was observed in CD8+ cytotoxic T lymphocytes that are cycling or exhausted (Fig. EV3G,H,K). Notably, melanoma tumor cells expressed low amounts of MIR155HG (Fig. EV3D,J), suggesting miRNAs not only indicate the presence of immune cells within the TME but can also mark their activation state.

Pre-existing immune infiltration and particularly IFNγ signaling have been linked to better clinical outcomes in cancer and improved responses to immunotherapies (Van Allen et al, 2015; Ayers et al, 2017). After observing the association of MIR155HG

with immune parameters within the TME, we next asked whether MIR155HG expression prior to ICI could stratify melanoma survival and therapy response. We separately categorized patients at the median MIR155HG for each of the five melanoma ICI data sets and, similar to what was observed for TCGA-SKCM, noted MIR155HG-high tumors showed an improved overall survival (Fig. 6A). When the patients were also segregated by immunotherapy response, we noted that the higher MIR155HG still marked better prognosis in non-responding patients, but not in the responding cohort (Fig. EV4A). We next wanted to examine the baseline differences at MIR155HG levels between responding and non-responding patients and noted that MIR155HG expression was elevated in one of the 5 melanoma ICI data sets prior to treatment with anti-PD1 or a combination of anti-CTLA4/anti-PD1 ICI (Fig. 6B) (Gide et al, 2019). Interestingly, ICI-responsive tumors tended to express higher levels of MIR155HG in this data set both prior to and during the immunotherapy although this was not statistically significant (Fig. EV4B). To understand whether miRNA expression can predict positive responses to ICI, we binarized therapy response (complete and partial responders vs progressive disease cases) and utilized binomial lasso regression in the pre-treatment combined melanoma data set. Our modeling revealed 11 and 17 miRNAs with positive and negative coefficients, respectively at the highest penalization level within 1 standard error of the maximum model AUC (Fig. EV4C–E). Interestingly, some of the miRNAs associated with ICI response in this analysis (including MIR155HG, MIR23AHG and MIR181A1HG) were also observed when using lasso regression against IFNγ response as the response variable. As melanoma molecular subtype can impact prognosis and therapy response (Jönsson et al, 2010; Hilke et al, 2020); we performed chi-square analysis to examine the relationship between the mutational subtypes and the immunotherapy response in the CRI-iAtlas melanoma data sets featured on TCGEx. We found no statistically significant associations between therapy response (R/NR) and the mutational subsets (BRAF, NRAS, NF1, WT) ($P = 0.2021$) (Appendix Fig. S4a–c). We then expanded the scope of the lasso regression and utilized the aggregated Pan-ICI data set as described above to investigate miRNAs as potential predictors of ICI response. This analysis returned only two miRNA genes with negative coefficients at the maximal penalization level (value of lambda within 1 standard deviation for the maximum cross-validated area under the roc curve [AUC]), however, relaxing the penalization level (value of

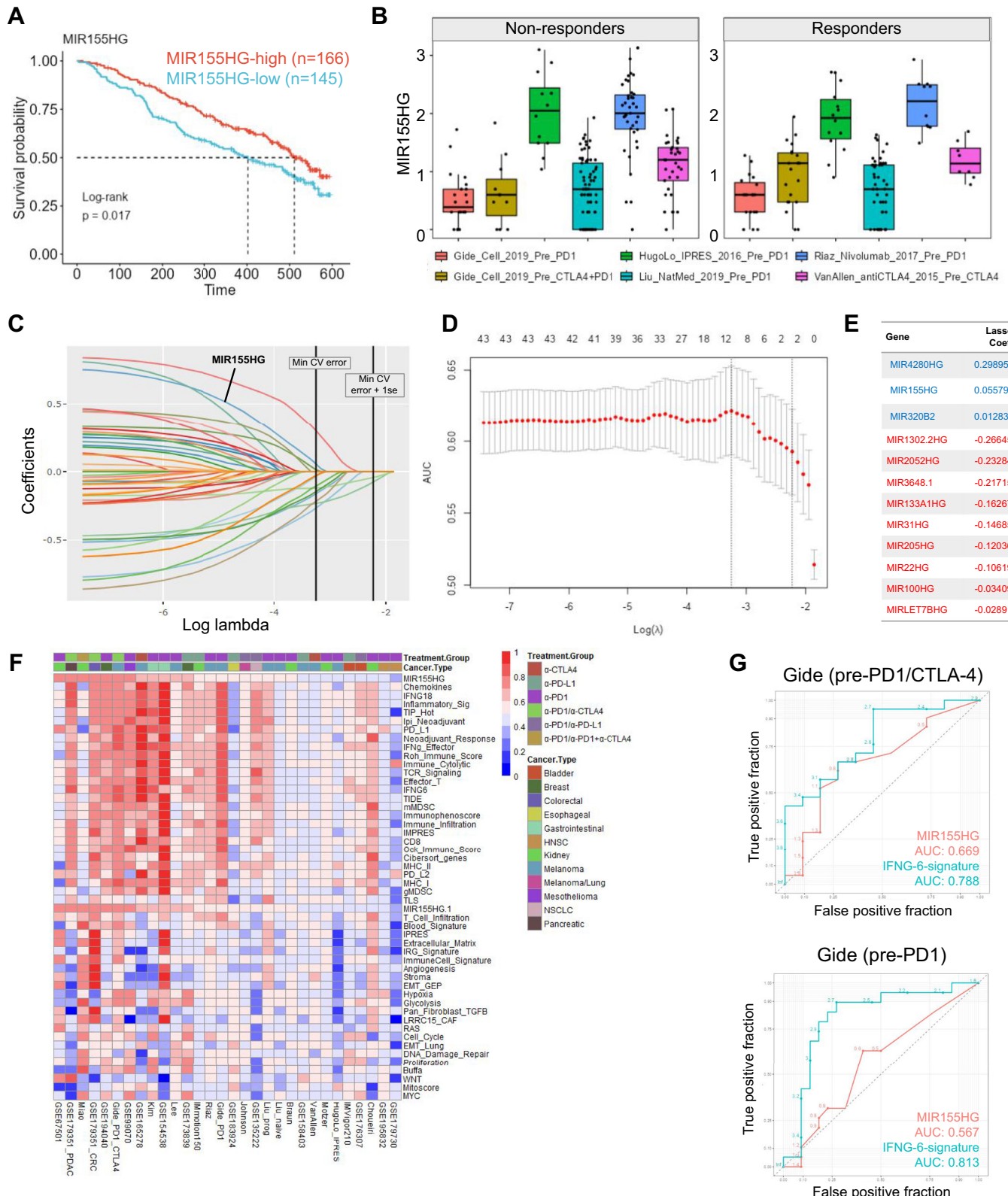

**Figure 6. Investigating the associations between miRNAs and immunotherapy response.**

(A) The higher MIR155HG expression in tumors prior to immune checkpoint inhibition (ICI) corresponds to improved survival in metastatic melanoma. (B) Boxplot shows MIR155HG expression in tumors responding or refractive to ICI agents. MIR155HG expression is higher in responders in some but not all studies examined (Gide Pre-PD1 $n = 41$, Gide Pre-CTLA4 + PD1 $n = 32$, Hugo Pre-PD1 $n = 26$, Liu Pre-PD1 $n = 121$, Riaz Pre-PD1 $n = 51$, Van Allen Pre-CTLA4 $n = 42$). The box represents the interquartile range (IQR), spanning from the 25th percentile (lower bound) to the 75th percentile (upper bound). The horizontal line within the box indicates the median (50th percentile). Whiskers extend to the smallest and largest values within 1.5× IQR from the lower (Q1) and upper (Q3) quartiles, respectively. (C–E) Binomial lasso regression was applied to pre-treatment SKCM immunotherapy studies ($n = 313$). In these analyses, a positive response (partial or complete response) is binarized to "1" and no response (progressive disease) is binarized to "0" as the response variable, and miRNA host gene transcripts in the RNAseq data were used as predictors. Regularized coefficients at various levels of lambda (C) and the AUC of the model (D) are shown. Error bars represent the minimum cross-validation error and minimum cross-validation error + 1 standard error lines. Nonzero coefficients were extracted at the lambda value with the highest cross-validated AUC (E). (F) Performance of MIR155HG and various other gene signatures for predicting ICI response is assessed by the IOSig platform and summarized via the heatmap. The color-coded AUC estimates between 0 and 1 from 12 different cancer clinical trials are shown. The heatmap was generated in the R environment using the data exported from the IOSig platform. (G) The ROC module of TCGEx was used to assess the potential of MIR155HG and 6-gene IFNγ signatures in predicting response to immunotherapy (complete/partial response vs progressive disease) in a data set generated by Gide et al.

lambda at the maximum cross-validated AUC) resulted in a total of 12 nonzero coefficients in the model (Fig. 6C–E). MIR155HG was among the 3 miRNAs with positive coefficients in the model, although the overall model performance was not too high when only using miRNAs as predictors. Although MIR155HG was not found to be a powerful predictor of response in these analyses, it was consistently expressed across many data sets examined for this study. Thus, we used a recently reported online portal, Immuno-Oncology Signatures Explorer (IOSig) (Coleman et al, 2023), to assess the association of MIR155HG with ICI responsiveness in a larger cohort of ICI studies in comparison to other previously published response signatures (Dataset EV2; Table EV2). We noted that the single-gene MIR155HG signature was variably associated with ICI response, although an AUC between 0.7 and 0.8 was seen for 8 out of 30 pre-ICI treatment data sets where MIR155HG expression was detected (Fig. 6F; Dataset EV3). Notably, these studies involved various cancer types including melanoma, kidney, breast, and gastrointestinal cancers. However, due to variation across data sets and small sample sizes, it is not straightforward to identify tumor types where MIR155HG expression consistently predicts better responses (Appendix Fig. S4d). We validated IOSig platform's findings using Gide et al data set in the ROC module of TCGEx and showed that MIR155HG expression could distinguish responders (complete and partial response) from non-responders (stable or progressive disease), albeit more poorly compared to a previously published 6-gene IFNγ response signature shown to predict immunotherapy response (Fig. 6G) (Ayers et al, 2017). Of note, MIR155HG expression performed better in distinguishing responders to the anti-PD1/anti-CTLA4 combination therapy in this data suggesting differences in molecular pathways that are involved in the mechanisms of ICI therapeutics. Thus, our results suggest that, while MIR155HG rises among other miRNAs to differentiate some ICI responders, it does not constitute a strong signature to be generalized across data sets.

Finally, we set out to identify a gene signature that can perform well across a broader range of ICI studies using the TCGEx analysis pipeline. As cytokine signaling is a core determinant of the immunological outcome in cancer, we utilized the 238-gene KEGG cytokine–cytokine receptor interaction gene set curated in the MSigDB repository (Liberzon et al, 2011). We binarized ICI response as described above and constructed a binomial elastic net

regression model (with elastic mixing parameter of alpha = 0.5) using the Pan-ICI pre-treatment data set described above. Out of the 238 genes, the calculated model retained 5 cytokines with positive coefficients, IFNG, CXCL9, CXCL13, IL21, and CCL5, at the highest regularization level (Fig. 7A–C). We next used the IOSig platform to assess the association of this signature (Cytokine-5) with the immunotherapy response across a diverse set of cancer types and treatments in addition to the TCGEx pan-cancer cohort. In these analyses, Cytokine-5 signature attained a higher overall AUC value compared to the previously published response signatures including 6- or 18-gene IFNγ response signature (Ayers et al, 2017) and Impres signature (Auslander et al, 2018), however, a degree of variation existed among data sets (Figs. 7D and EV5A,B; Dataset EV4). It is important to note that some of these signatures are debated in the field for broad applicability (Carter et al, 2019; Auslander et al, 2019), however, they are still included in our comprehensive analyses. Out of 34 pre-treatment data sets on IOSig platform expressing all 5 genes of the signature (including the data sets featured on TCGEx), 4 showed AUC values of 0.9 or more and, interestingly, these studies were conducted on 4 different cancer types (thymic, colorectal, non-small cell lung cancer and melanoma) and involved different treatment agents (anti-PD1, anti-CTLA4, or combination). In addition, the Cytokine-5 signature corresponded to an AUC between 0.7 and 0.9 in 8 other data sets, and an AUC between 0.6 and 0.7 for yet another 8 data sets across cancer types and treatment modalities. We then wanted to examine the AUC scores of Cytokine-5 signature in the IOSig data sets after excluding the ones that are featured as pan-cancer data on the TCGEx platform to serve as an independent validation (a total of 22 studies annotated in the IOSig database). Out of these studies, 4 had an AUC of 0.9 or higher, and another 6 had an AUC value between 0.6 and 0.9. Furthermore, our analyses revealed that, among the previously published signatures, Cytokine-5 had a higher average AUC score for predicting immunotherapy responsiveness, although the differences were not too striking between the highest-scoring signatures (Fig. 7E; Dataset EV4). Notably, the 6-gene IFNg signature previously reported to predict anti-PD-1 response (Ayers et al, 2017) showed a positive correlation with the Cytokine-5 signature, although the Cytokine-5 score performed better in predicting responses in some cancer types (Fig. 7F and EV5B). It is important to note that a previously reported 12-gene chemokine signature (marked as Chemokines in the

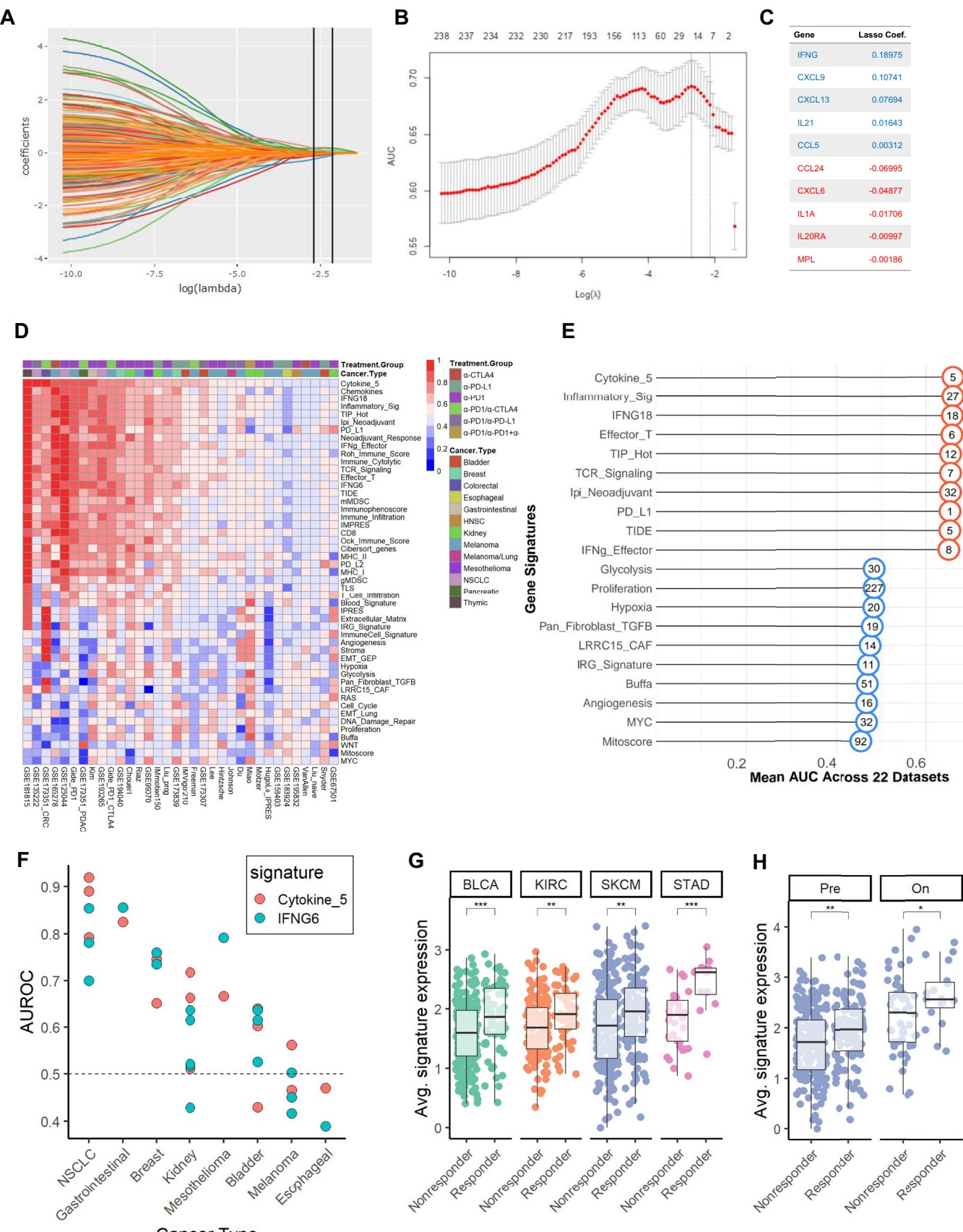

**Figure 7. TCGEx identifies a new cytokine signature predicting immunotherapy response in a broad range of cancers.**

(A–C) Binomial elastic net regression (alpha = 0.5) was performed using the Pan-ICI data consisting of 938 samples across 10 clinical trials and 4 different tumor types. In these analyses, binarized ICI response (CR/PR vs PD) was used as the independent variable, and 238 transcripts associated with cytokine–cytokine receptor interaction (KEGG) were used as independent (i.e., predictor) variables. Regularized coefficients (A), and the AUC of the cross-validated model (B) are shown and the error bars indicate the range of lambda values within one standard error of the maximum AUC. Predictors with nonzero coefficients at the smallest value of lambda within the 1 standard error of the maximum AUC of the cross-validated model are listed (C). "Cytokine-5" signature is derived from the transcripts with positive coefficients. (D) The IOSig platform was used to assess the potential of Cytokine-5 and other previously published signatures for predicting immunotherapy response. AUC values ranging between 0 and 1 from various studies are depicted with the color gradient. The graph was generated in the R environment using the data exported from the IOSig platform. (E) The average AUC of the top and bottom signatures are summarized in the graph where the numbers in the circles indicate the number of genes in the signature. (F) The AUC values of Cytokine-5 and IFNG6 signatures in individual clinical trials are shown across various cancer types. Each point indicates a distinct data set (data were obtained from the IOSig platform). (G) Boxplot module of TCGEx was used to plot the average expression of Cytokine-5 signature in tumors responding or refractory to ICI agents. Individual samples collected prior to ICI treatment are depicted by points (BLCA $n = 348$, KIRC $n = 207$, SKCM $n = 313$, STAD $n = 45$). The box represents the interquartile range (IQR), spanning from the 25th percentile (lower bound) to the 75th percentile (upper bound). The horizontal line within the box indicates the median (50th percentile). Whiskers extend to the smallest and largest values within 1.5× IQR from the lower (Q1) and upper (Q3) quartiles, respectively. Statistical comparisons are performed using the $t$ test, and the $P$ values are adjusted for multiple comparisons (***$P < 0.001$, **$P < 0.01$). (H) Boxplots using the data from melanoma ICI studies demonstrate responder tumors express higher levels of Cytokine-5 signature prior to and during the treatment (Pre $n = 313$, On $n = 76$). The box represents the interquartile range (IQR), spanning from the 25th percentile (lower bound) to the 75th percentile (upper bound). The horizontal line within the box indicates the median (50th percentile). Whiskers extend to the smallest and largest values within 1.5× IQR from the lower (Q1) and upper (Q3) quartiles, respectively. Statistical comparisons are performed using the $t$ test and the $p$ values are adjusted for multiple comparisons (**$P < 0.01$, *$P < 0.05$).

heatmap) containing CCL5 and CXCL9 had a lower albeit comparable overall performance in predicting ICI response. Similarly, the 6- or 18-gene IFNγ signature described by Ayers et al (Ayers et al, 2017) (IFNG6 or IFNG18) demonstrated a strong positive correlation with the Cytokine-5 signature, but they had a comparable but slightly lower prediction power overall. Notably, two genes from Cytokine-5 signature, IFNG and CXCL9, overlapped with the IFNG6 gene set, while CCL5 and CXC13 genes were shared with IFNG18. Importantly, these results do not assert the superiority of the Cytokine-5 signature, which would require further computational and experimental validation in other data sets; instead, they demonstrate the capability of TCGEx in identifying biomarkers that can be applicable to a wider context. Our findings indicate that the Cytokine-5 signature—which has partial overlaps with some of the other signatures—could perform better than or comparably to previously defined predictors in multiple data sets. To further visualize the distribution and range of this signature, we also examined its average expression in the pre-ICI data sets available on the TCGEx platform. As expected, the Cytokine-5 signature was found significantly higher in responders at baseline across melanoma, kidney, stomach, and bladder cancers (Figs. 7g and EV5C). Interestingly, the higher levels of this signature in therapy-responsive melanoma samples were evident both prior to and on ICI treatment. We also noted that the expression of this signature increased during ICI treatment regardless of the response classification suggesting that the Cytokine-5 signature is induced upon immunotherapy, albeit more strongly in responding tumors (Fig. 7H). Lastly, we examined how Cytokine-5 signature performs in specific cancers and treatment types. Although the number of studies was limiting for some cancers, Cytokine-5 score corresponded to an AUC of >0.75 for multiple cancer types including thymic, colorectal, pancreatic, gastrointestinal, and non-small cell lung cancers (Appendix Fig. S5a). When we categorized studies based on treatment type, we observed that Cytokine-5 signature has an AUC of >0.6 for single agent anti-CTLA4 or anti-PD1 as well as their combinations, although these patterns may be confounded by cancer type (Appendix Fig. S5b). Collectively, these results highlight the utility

of the TCGEx web server to investigate cancer transcriptomics data and identify clinically relevant predictors such as the Cytokine-5 signature in the context of immunotherapy.

## Discussion

TGCEx platform provides a user-friendly solution for high-throughput data analysis and can benefit researchers beyond the scope of this work. Although we developed this tool with RNAseq/miRNAseq/proteomics data in mind and not tested elsewhere, the TCGEx modules are also compatible with other data types, such as microarray, Nanostring, and CHIP-seq data. While this platform allows users to upload and explore their own data through flexible interfaces, it is important to consider the effects of data normalization and filtering on the analyses. We provide convenient checks and input options to help with data integration and processing, but more customized data manipulations may be needed in specific study contexts. We openly provide the TCGEx source code to help researchers who wish to further control the analysis pipeline and tailor the code to their needs.

Taken together, TCGEx provides a powerful and flexible interface for analyzing cancer transcriptomics data originating from various sources. The TCGEx pipeline can be adapted to study various aspects of cancer including oncogenic drivers, immune associations, prognostic markers in specific subtypes, among others. Furthermore, TCGEx allows integrative analyses of multiple data sets to study the molecular underpinnings of human cancer and enable users to interact with their own data. Visual analysis interfaces increase the accessibility and reusability of high-throughput data and facilitate research by serving as a bridge between scientific disciplines. In the future, we will test the TCGEx pipelines with other types of data, expand the preprocessed data selection, and actively monitor the GitHub repository to engage with the users. TCGEx, publicly and freely accessible on the web, not only reduces barriers to data analysis for researchers with no programming experience but also can be implemented into larger analysis pipelines to facilitate hypothesis-driven research.

# Methods

## Reagents and Tools Table

| Reagent/resource | Reference or source | Identifier or catalog number |
|---|---|---|
| **Experimental models** | | |
| **Recombinant DNA** | | |
| **Antibodies** | | |
| **Oligonucleotides and other sequence-based reagents** | | |
| **Chemicals, enzymes, and other reagents** | | |
| **Software** | | **Packages** |
| R-Programming and R-Studio (R version 4.4.1) | https://www.R-project.org/. R Core Team (2013) | edgeR 4.2.2<br>limma 3.60.6<br>corrplot 0.95<br>Hmisc 5.2-1<br>openxlsx 4.2.7.1<br>plyr 1.8.9<br>gridExtra 2.3<br>fgsea 1.30.0<br>shinyBS 0.61.1<br>janitor 2.2.1<br>kableExtra 1.4.0<br>factoextra 1.0.7<br>survival 3.8-3<br>survminer 0.5.0<br>colourpicker 1.3.0<br>shinyvalidate 0.1.3<br>extrafont 0.19<br>ggiraph 0.8.11<br>broom 1.0.7<br>plotROC 2.3.1<br>shinyalert 3.1.0<br>shinybusy 0.3.3<br>fresh 0.2.1<br>shinydashboardPlus 2.0.5<br>shinyjs 2.1.0<br>DT 0.33<br>shinythemes 1.2.0<br>glmnet 4.1-8<br>Matrix 1.7-1<br>lubridate 1.9.4<br>forcats 1.0.0<br>stringr 1.5.1<br>purrr 1.0.2<br>readr 2.1.5<br>tidyr 1.3.1<br>tibble 3.2.1<br>tidyverse 2.0.0<br>rintrojs 0.3.4<br>heatmaply 1.5.0<br>viridis 0.6.5<br>viridisLite 0.4.2<br>msigdbr 7.5.1<br>RColorBrewer 1.1-3<br>pheatmap 1.0.12<br>readxl 1.4.3<br>shinyWidgets 0.8.7<br>dplyr 1.1.4<br>plotly 4.10.4<br>data.table 1.16.4<br>ggpubr 0.6.0<br>ggplot2 3.5.1<br>shinydashboard 0.7.2<br>shiny 1.10.0 |
| Docker | https://www.docker.com/ Merkel D, 2014 | – |
| GitHub | https://github.com/ | – |
| **Other** | | |

## Summary of the pipeline

TCGEx was created using the R programming language and Shiny framework and it runs on a publicly accessible GNU/Linux server. TCGEx is designed to work with preprocessed harmonized data allowing rapid and interactive analyses of TCGA projects, landmark ICI studies, and custom data sets. Preprocessed TCGA data for the pipeline were prepared by integrating RNA sequencing (RNAseq), microRNA sequencing (miRNAseq), reverse phase protein array (RPPA) proteomics data, and clinical metadata. To that end, "tidy" gene expression data frames were merged by the 16-digit TCGA sample barcodes followed by the addition of clinical metadata at the patient level by matching the 12-digit TCGA sample barcode. The latter data type contains dozens of features ranging from patient demographics to tumor subtypes and genetic aberrations, and it changes depending on the study. We further expanded the sample-level metadata by incorporating intratumoral immune cell signatures described by Thorsson et al (Thorsson et al, 2018). The ICI data sets were downloaded as is from cBioPortal and CRI-iAtlas repositories (Gao et al, 2013; Eddy et al, 2020), and prepared for the TCGEx pipeline. cBioPortal data sets had variable normalization methods including CPM, RPKM, and FPKM; whereas CRI-iAtlas data sets were quartile-normalized. These data sets were used directly as hosted in the source repositories, with a few minor adjustments to the variable names to implement in our platform (the code for preparing these data sets can be found on the TCGEx GitHub repository). TCGEx provides convenient access to the entire CRI-iAtlas data object (i.e., pan-cancer) and its subsets divided based on the specific study. In addition, the sample-level metadata from these repositories were used directly without altering variable names; thus selection of multiple data sets for integrative analyses will only keep the variables that are shared by these data sets. TCGEx analysis starts by selecting one or more cancer projects in the data selection module. After data selection, relevant data objects are loaded into the session and passed to individual analysis modules (Fig. 1). TCGEx offers the following user-friendly modules with graphical interfaces: (i) Kaplan–Meier survival analysis; (ii) Cox proportional hazards survival analysis; (iii) feature-to-metadata exploratory graphing; (iv) feature-to-feature correlation graphing; (v) correlative features analysis; (vi) heatmap and hierarchical clustering; (vii) gene set enrichment analysis (GSEA); (viii) receiving operating characteristics (ROC) analysis; (ix) principal component analysis (PCA); and (x) lasso/ridge/elastic net machine learning analysis. Each module allows users to examine specific data subsets such as metastatic and/or primary tumor biopsies as well as healthy control samples. Furthermore, flexible options enable users to tailor the pipeline to their specific needs and generate informative publication-ready plots. The TCGEx Shiny app consists of modularized R scripts facilitating troubleshooting and incorporation of the code into other analysis pipelines. Thus, TCGEx aims to reduce barriers to high-throughput data analysis in cancer research by providing a scalable and user-friendly solution.

## Data selection module

This module provides the interface for selecting from 33 TCGA and 14 ICI data sets to be analyzed in TCGEx. Upon data selection,

preprocessed data objects are loaded, and descriptive statistics are visualized to help users examine the age and sex distribution of the selected patients as well as the tissue origin of the biopsy samples. TCGA data objects were prepared by the sample-level merging of the log10(cpm+1) transformed RNAseq/miRNAseq data, normalized RPPA proteomic data, and clinical metadata. To facilitate downstream analyses, we removed genes where expression was zero or not available (NA) in 25% or more of the samples in the TCGA data set. ICI studies obtained from CRI-iAtlas repository were quartile-normalized while the data downloaded from cBioPortal were normalized through RPKM, FPKM, or TPM methods. For convenience and more control over the analyses, we added options for normalizing user-uploaded data sets and filtering lowly-expressed genes. Unlike some of the previously available tools for TCGA data analysis (Goldman et al, 2020; Tang et al, 2019), TCGEx allows users to select single or multiple data sets for integrative analyses. It is important to note that data sets normalized with different methods cannot be merged for integrated analyses. The data selection module prevents selection of incompatible data sets featured on TCGEx. Additionally, the same type of count data (such as TPM or FPKM) can be associated with technical variation depending on the software and the reference transcriptome used. Thus, although we took measures to minimize erroneous analyses, it is ultimately users' responsibility to ensure that the integration of data sets are appropriate. If multiple TCGA projects are selected, preprocessed data sets are aggregated keeping only the matched features and clinical metadata variables. In the end, the data selection module creates a tidy data frame containing normalized RNAseq/miRNAseq/RPPA data and clinical metadata; and feeds it to the other analysis modules.

## Survival analysis modules

Studying the association between gene expression and survival outcome is important for understanding key mechanisms of cancer progression (Liu et al, 2018). TCGEx features two distinct survival analysis methods: Kaplan–Meier (KM) and Cox proportional hazards (CoxPH) modeling (Rich et al, 2010; Breslow, 1975). In both modules, categorical and continuous variables can be examined. Categorical meta-variables can differ between data sets, but it can include tumor grade, mutation types, and patient demographic information, among other features. Continuous variables such as gene expression and intratumoral immune cell signatures (Thorsson et al, 2018) can be directly modeled in CoxPH analysis or they can be categorized as "high", "mid", and "low" at user-defined thresholds for KM analysis. If more than one cancer type is selected by the user, gene expression can be categorized across all the projects at a common threshold, or within each cancer type separately. In KM analysis involving categorical variables, the user can further determine data subsets to include in the analysis and further customize the pipeline. Both KM and CoxPH modules allow including covariates in the model to examine the association of multiple variables with the survival outcome simultaneously. Furthermore, the CoxPH module is designed to accept user-defined interaction terms to accommodate arbitrarily complex multi-variable study designs. Survival analysis pipelines of TCGEx generate customizable plots and show various statistical details including log-rank *P* value for KM and CoxPH, and likelihood ratio test and Wald test p values along with hazard ratios for CoxPH. In addition, the proportionality assumption of the CoxPH model is checked using Schoenfeld residuals against time, and the user is prompted a message regarding the validity of the model. Taken together, TCGEx survival analysis modules provide flexible options for customizing analysis to specific needs and studying the relationship between gene expression and survival outcomes in cancer.

## Boxplot module

In this module, users can explore the relationship between gene expression and categorical clinical meta-variables. Categorical variables vary according to tumor type, and they include disease pathological stage, patient sex and race, tumor molecular subtype (Berger et al, 2018; Cancer Genome Atlas Network, 2015), genome-wide methylation state (Cancer Genome Atlas Network, 2015), oncogenic signaling (OncoSign) cluster type (Ciriello et al, 2013), genetic aberrations (ICGC/TCGA Pan-Cancer Analysis of Whole Genomes Consortium, 2020), genomic instability levels (Muzny et al, 2012), intratumoral lymphocyte scores (Cancer Genome Atlas Network, 2015), and numerous other metadata features (Dataset EV1). This module allows selecting specific categories to generate gene expression plots to emphasize differences in data subsets. To offer more flexibility in the analyses, we also added an optional faceting variable which can further break data into subcategories. This way, for instance, it becomes possible to plot the expression of a given gene in various mutation subsets in male and female patients separately. To facilitate publication-ready plotting, the program allows the user to select journal-specific color palettes, toggle individual data points on/off, and define specific statistical comparisons to highlight. Here, users can select the t test or Wilcoxon test for pairwise comparisons between data sub-groups. Thus, this module helps researchers explore gene expression in clinically meaningful data subsets.

## Correlation analysis modules

Examining correlations among genes and continuous metadata features can help study co-regulation mechanisms (Van Allen et al, 2015) and miRNA-mRNA interactions (Van der Auwera et al, 2010). TCGEx offers two modules for correlation analysis which can generate sample-level scatter plots or correlation matrices and tables. Sample-level plots are generated by plotting two continuous variables against each other, where each point denotes a patient sample. Continuous variables for this analysis can include transcript counts in RNAseq/miRNAseq data, protein expression data, and numerical metadata such as mutation load or intratumoral immune cell scores (Thorsson et al, 2018). The correlation scatter plot is designed to be responsive, and it can show patient barcode, sex, and race upon hovering over individual data points. Furthermore, an optional faceting variable can be used to break data points into various subcategories and generate informative plots, as described in the previous modules. Users can visualize the best-fitting line and the linear regression equation on the graph to demonstrate the relationship between variables. In addition to these functionalities, we anticipated that it could be of interest to examine the top positive and negative correlators of a specific gene and plot a gene-to-gene correlation matrix. The second correlation analysis module of TCGEx enables these analyses by calculating pairwise

Pearson and Spearman correlation coefficients and creating tabular and graphical outputs. Therefore, these pipelines facilitate examining the linear relationships between variables and help investigate potential genetic interactions.

## Heatmap and hierarchical clustering module

Heatmaps are commonly used for visualizing high-throughput gene expression data. Especially when combined with hierarchical clustering, heatmaps can reveal interesting patterns in the data and help categorize samples with different characteristics (Sørlie et al, 2001). We created an interactive heatmap module equipped with flexible options to facilitate visualization and analysis of transcriptomics data. Here, users can manually enter genes of interest, or select curated gene sets described in the Molecular Signatures Database (MSigDB) (Liberzon et al, 2011). Extensively utilized in both cancer and non-cancer literature, MSigDB contains thousands of annotated gene sets that define numerous biological states and processes. We anticipated that heatmaps created with these predefined gene sets could be especially informative when paired with user-selected sample-level annotations. Thus, we added functionality to show custom annotation bars above the heatmaps. In these annotation bars, both categorical and continuous features can be specified to generate complex heatmaps. As described previously, numerous cancer-specific categorical features are available for selection including tumor stage, mutational subtype, and oncogenic copy number alterations (Carter et al, 2012). When continuous annotation variables are selected, we designed the pipeline to categorize the samples at the median value to create two color-coded groups. The pipeline also allows selecting multiple continuous variables, in which case the categorization is done for each variable separately or after averaging the variables, thus making it possible to create meta-features on the fly. In the heatmap module, users can filter out genes with low variance and change the hierarchical clustering parameters to tailor the analysis to specific needs. Therefore, this module can help researchers create informative heatmaps to highlight meaningful patterns in gene expression data.

## Gene set enrichment analysis module

In transcriptome analysis, expression patterns of genes in specific pathways can be examined to study the biological states of the samples. Gene set enrichment analysis (GSEA) is a powerful approach for comparing two groups of samples by focusing on a list of genes that share common biological characteristics (Subramanian et al, 2005). We developed an efficient GSEA pipeline that allows comparisons between user-defined data subsets. Similar to previous modules, users can define these data subsets by selecting specific values for categorical variables (such as mutation subtypes, tumor stage, and patient sex), or by binarizing continuous variables (such as gene expression, non-silent mutation rate, and intratumoral immune scores) at custom thresholds. After defining two data subsets, the pipeline ranks genes based on their expression levels and variance using the previously defined signal-to-noise approach (Subramanian et al, 2005). The GSEA module utilizes thousands of readily available gene sets from the MSigDB repository and facilitates the investigation of various biological pathways in iterative analyses (Liberzon et al, 2011). We further

added gene signatures predictive of immunotherapy response in numerous studies that are curated in a recent report (Coleman et al, 2023). Importantly, to further increase the usability of this module, we gave the user an option to provide custom gene sets and examine their expression patterns in user-defined data subsets. This module can create enrichment plots for individual gene sets and print a table of leading-edge genes that mostly drive the enrichment scores (Subramanian et al, 2005), or it can show the most highly enriched gene sets among others provided to the pipeline. Taken together, the GSEA module provides a flexible and customizable platform for examining functional associations in cancer gene expression data.

## Receiving operator characteristics analysis module

Receiving operator characteristics (ROC) analysis is a method for evaluating the discriminatory performance of variables in a binary classification system (Metz, 1978). ROC curves are generated by plotting the true positive rate (i.e., sensitivity) against the false positive rate (i.e., 1-specificity) across all possible thresholds for the variable of interest. The area under the ROC curve (AUC or AUROC) can be examined to assess the power and accuracy of the classifier, a commonly used approach in the field for investigating diagnostic and prognostic biomarkers (Kreft et al, 1997; Guo et al, 2019; Lim et al, 2019). TCGEx features a powerful and flexible module for performing ROC analyses using gene expression data and clinical metadata. Users can binarize features of interest at custom thresholds to create two groups needed for the ROC analysis and specify custom predictor variables to assess their classifier potential. Using this pipeline, one can examine, for instance, whether the expression of a specific gene is associated with certain tumor characteristics such as mutation rate, and intratumoral immune signatures. As in the previously described modules, users can work with both categorical variables and numerical variables through flexible input options and tailor the analysis to specific needs. We also provided an option to add ROC curves to the graph generated using MSigDB gene sets. This design facilitates comparing user-selected custom predictors and other functionally annotated predictors. Thus, we anticipate that the ROC module can be conveniently used in various study contexts to assess novel predictors in cancer data.

## Principal component analysis module

Dimensionality reduction techniques such as principal component analysis (PCA) are commonly employed in transcriptomics (Yeung and Ruzzo, 2001). PCA involves constructing a new coordinate system for the data using linear combinations of its variables (i.e., genes in RNAseq). The axes of this new coordinate system, called principal components, represent the variation in the experiment, and they help visualize multidimensional data sets on a 2D graph. TCGEx program features a user-friendly PCA module with flexible inputs to be utilized in various study contexts. Similar to other modules, one can specify sample types and select a custom list of genes to prepare the gene expression matrix for PCA. Here, the module conveniently allows selecting all genes in the RNAseq/miRNAseq data or specifying genes from MSigDB pathways. Furthermore, convenient options allow selecting miRNAs or lncRNAs exclusively to tailor analyses to specific contexts.

Although PCA accounts for both highly and lowly variable genes, we provided a pre-filtering option to speed up the analysis by removing genes with low variance. In PCA plots, it is usually helpful to annotate individual points based on specific characteristics. For instance, one may want to see whether samples with certain genetic features form a separate cluster in the PCA space (Park et al, 2018). To address this need, this module allows color-coding data points based on categorical or continuous variables, the latter of which is categorized at the median value to form "high" and "low" groups. PCA module generates customizable and interactive graphics and allows exporting publication-ready plots.

### Machine learning module

Machine learning (ML) refers to a range of applications and algorithms that aim to extract relevant and useful information from data. In the context of bioinformatics, ML is utilized to perform classification, prediction, and feature selection from biological data (Shastry and Sanjay, 2020; Xu and Jackson, 2019). Regularized regression, a type of supervised machine learning technique, is a derivative of linear regression that allows one to simultaneously create a model and perform feature selection in high dimensional data. It is optimized to minimize the sum of squared residuals while penalizing the generated model coefficient estimates. The penalty term is applied to the model equation to reduce model complexity and make the prediction with the limited mean-squared error. Ridge, lasso, and elastic net regression are all types of regularized regression methods with varying strengths depending on the underlying structure of the data. Ridge regression shrinks the estimated coefficients without making them zero and assigns correlating parameters with similar coefficients (Friedman et al, 2010). Lasso regression on the other hand can shrink model coefficients to zero and therefore performs feature selection (Tibshirani, 1996). Elastic net regression can refer to a middle ground in between ridge and lasso regression and it is ideal for data sets in which the number of predictor variables significantly exceeds the number of samples and/or there is a high number of correlating variables. The optimal degree of penalization for each approach can be selected using cross-validation across the data set. The ML module in TCGEx provides a user-friendly interface for applying these regularized linear regression methods on the transcriptomics data. Users choose response and predictor variables for the model and specify the ML algorithm by setting the penalization parameter alpha. The response variable (i.e., dependent variable) in these analyses can be constructed from a custom list of genes or MSigDB gene sets by taking the average of the expression values. Alternatively, users can select categorical metadata features as the response variable and binarize data subsets in a custom way to adapt the pipeline to various study contexts. This module also provides convenient options to specify different types of predictor variables (i.e., independent variables) including genes belonging to specific biological pathways or encoding miRNAs and lncRNAs. This way, one can easily examine, for instance, miRNA/lncRNAs that are most closely associated with a pathway of interest (Lu et al, 2011). The ML module can use the entire data to train a model or split it into training and testing subsets to evaluate the overall model accuracy (Xu and Jackson, 2019). The findings of the analyses are displayed through interactive graphs which show penalized coefficients and the

mean-squared error of the cross-validated model across various levels of regularization. Taken together, the ML module provides a customizable platform for sophisticated analysis of gene expression data.

## Data availability

The TCGEx web server is accessible at https://tcgex.iyte.edu.tr. All code for the TCGEx app, data download, and pre-processing are publicly accessible at https://github.com/atakanekiz/TCGEx. To facilitate local execution and code development, TCGEx docker image can be accessed at https://hub.docker.com/repository/docker/atakanekiz/tcgex/. The data sets used in TCGEx were derived from the TCGA data repository in the public domain (https://portal.gdc.cancer.gov/). Processed data files and TCGEx source code are accessible on Figshare (https://doi.org/10.6084/m9.figshare.23912532) (https://figshare.com/s/22f21b780acd57fb5dfb).

The source data of this paper are collected in the following database record: biostudies:S-SCDT-10_1038-S44319-025-00407-7.

## Peer review information

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

## Acknowledgements

The authors thank the interim director of the IzTech IT Department Dr. Ozgur Orun for helpful discussions in hosting the TCGEx app and facilitating the establishment of the Shiny server at IzTech. The authors also extend our gratitude to Emin Bayindirli for his involvement during the initial development of the app. The authors are also grateful to the TCGA initiative as a whole and the patients who participated in the TCGA projects and clinical trials. Funding: AA received Abdi Ibrahim Foundation undergraduate scholarship. CS, EK, and MMO received Turkish Health Institutes Directorate (TUSEB) undergraduate research project funding (TUSEB-A1-28154). CS and EK also received Scientific

and Technological Council of Turkey (TUBITAK) 2247C-STAR scholarships. MEK was supported by a TUBITAK-2210-A graduate scholarship. HAE is supported by EMBO (IG-5714-2024), TUBITAK (2232-121C115, 1001-122S337), Turkish Academy of Sciences (TUBA-GEBIP-2022), and institutional grants (2022IYTE-2-0060, 2023IYTE-1-0053, 2023IYTE-1-0054).

## Author contributions

**M Emre Kus**: Conceptualization; Resources; Data curation; Software; Formal analysis; Validation; Investigation; Visualization; Methodology; Writing—original draft; Writing—review and editing. **Cagatay Sahin**: Conceptualization; Data curation; Software; Formal analysis; Validation; Investigation; Visualization; Methodology; Writing—original draft; Writing—review and editing. **Emre Kilic**: Conceptualization; Data curation; Software; Formal analysis; Validation; Investigation; Visualization; Methodology; Writing—original draft; Writing—review and editing. **Arda Askin**: Conceptualization; Data curation; Software; Formal analysis; Validation; Investigation; Visualization; Methodology; Writing—original draft; Writing—review and editing. **M Mert Ozgur**: Conceptualization; Data curation; Software; Formal analysis; Validation; Investigation; Visualization; Methodology; Writing—original draft; Writing—review and editing. **Gokhan Karahanogullari**: Software; Formal analysis; Writing—original draft. **Ahmet Aksit**: Conceptualization; Resources; Software; Methodology; Writing—original draft; Writing—review and editing; Establishment of the distributed TCGEx server. **Ryan M O'Connell**: Resources; Formal analysis; Investigation; Writing—original draft; Key input and support during early concept development and prototyping. **H Atakan Ekiz**: Conceptualization; Resources; Data curation; Software; Formal analysis; Supervision; Funding acquisition; Validation; Investigation; Visualization; Methodology; Writing—original draft; Project administration; Writing—review and editing.

Source data underlying figure panels in this paper may have individual authorship assigned. Where available, figure panel/source data authorship is listed in the following database record: biostudies:S-SCDT-10_1038-S44319-025-00407-7.

## Disclosure and competing interests statement

The authors declare no competing interests.

# Expanded View Figures

**Figure EV1.   Expression profiles of subtype-selective genes in melanoma (related to Fig. 2).**

(A) Expression of selected genes demonstrates comparable patterns between metastatic (TM) and primary (TP) tumor samples in SKCM (TM $n = 368$, TP $n = 103$, BRAF mutants $n = 146$, NF1 mutants $n = 27$, RAS mutants $n = 91$, Triple WT $n = 44$). (B, C) Expression of previously reported genes distinguishing melanoma mutational subtypes may show sex-specific expression patterns (female $n = 180$, male $n = 291$, BRAF mutants $n = 146$, NF1 mutants $n = 27$, RAS mutants $n = 91$, Triple WT $n = 44$). (D) Other transcripts selectively expressed in mutational subtypes of melanoma are shown (BRAF mutants $n = 146$, NF1 mutants $n = 27$, RAS mutants $n = 91$, Triple WT $n = 44$). The box represents the interquartile range (IQR), spanning from the 25th percentile (lower bound) to the 75th percentile (upper bound). The horizontal line within the box indicates the median (50th percentile). Whiskers extend to the smallest and largest values within 1.5× IQR from the lower (Q1) and upper (Q3) quartiles, respectively. $T$ test $P$ values are adjusted for multiple comparisons ($P < 0.0001$, "****"; $P < 0.001$, "***"; $P < 0.01$, "**"; $P < 0.05$, "*"; $P > 0.05$, "ns").

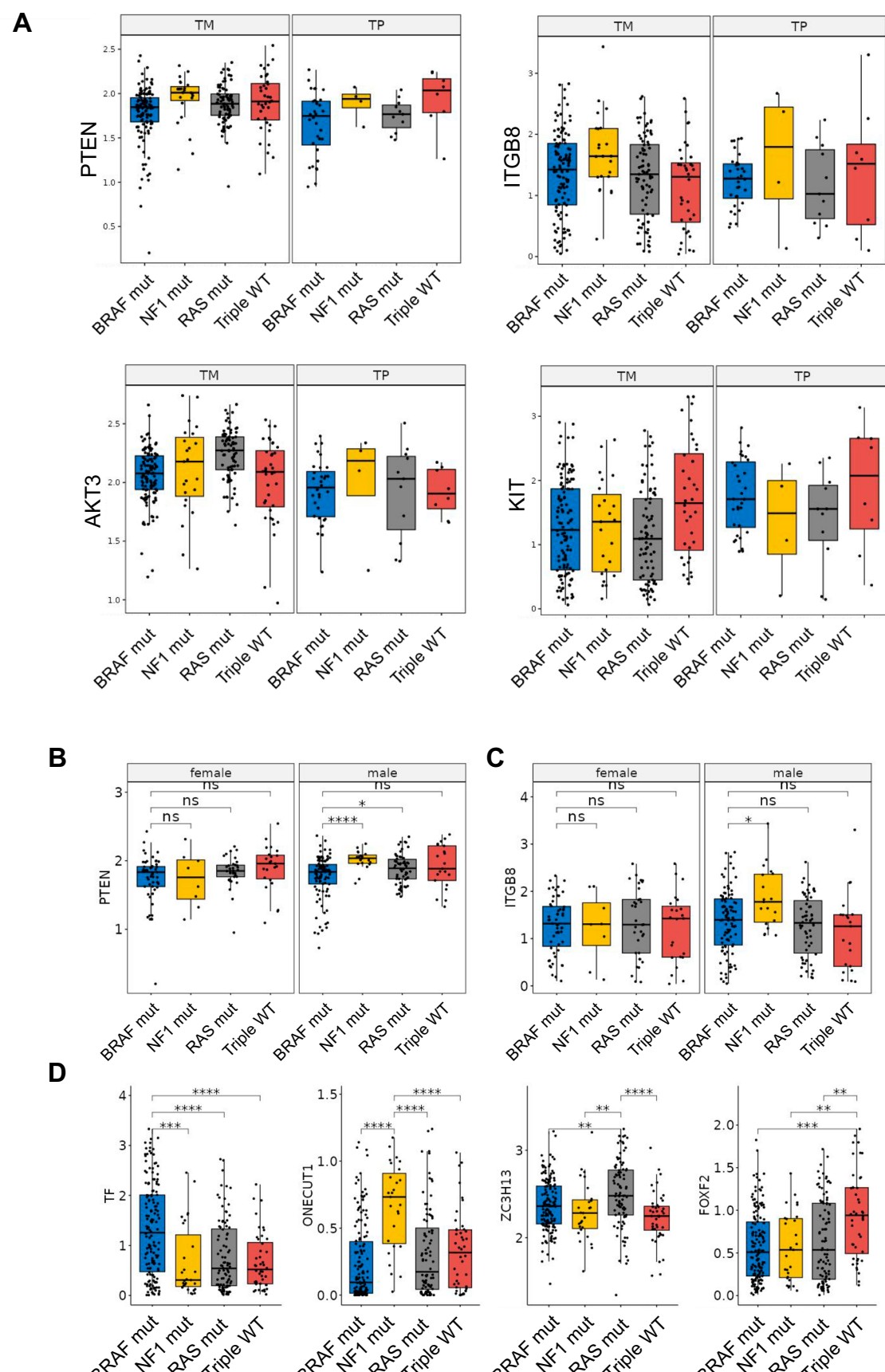

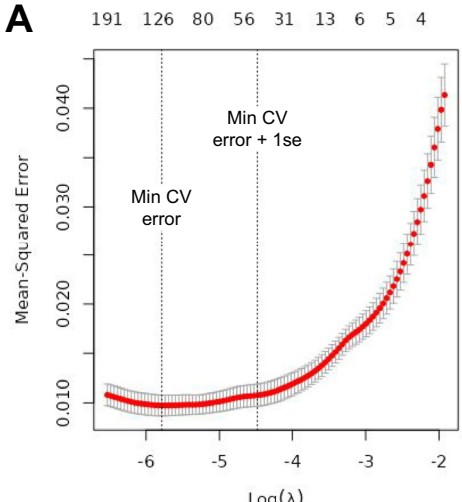

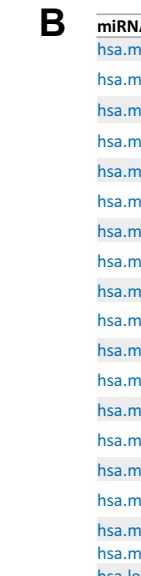

**B**

| miRNA | Coefficient |
| --- | --- |
| hsa.miR.7702 | 0.0792 |
| hsa.miR.3614.5p | 0.0719 |
| hsa.miR.155.5p | 0.0512 |
| hsa.miR.142.3p | 0.0469 |
| hsa.miR.5586.5p | 0.0324 |
| hsa.miR.181a.3p | 0.0307 |
| hsa.miR.100.5p | 0.0266 |
| hsa.miR.6125 | 0.0262 |
| hsa.miR.503.5p | 0.0260 |
| hsa.miR.150.5p | 0.0246 |
| hsa.miR.29b.2.5p | 0.0245 |
| hsa.miR.150.3p | 0.0203 |
| hsa.miR.342.5p | 0.0186 |
| hsa.miR.4728.3p | 0.0162 |
| hsa.miR.3614.3p | 0.0119 |
| hsa.miR.181c.5p | 0.0100 |
| hsa.miR.29c.5p | 0.0094 |
| hsa.miR.342.3p | 0.0080 |
| hsa.let.7b.3p | 0.0069 |
| hsa.miR.1228.3p | 0.0067 |
| hsa.miR.542.5p | 0.0057 |
| hsa.miR.598.3p | 0.0048 |
| hsa.miR.424.3p | 0.0039 |
| hsa.miR.21.3p | 0.0026 |
| hsa.miR.30a.5p | 0.0024 |
| hsa.miR.29c.3p | 0.0017 |

| miRNA | Coefficient |
| --- | --- |
| hsa.miR.3170 | -0.0360 |
| hsa.miR.6509.5p | -0.0235 |
| hsa.miR.4636 | -0.0204 |
| hsa.miR.3680.3p | -0.0191 |
| hsa.miR.92a.3p | -0.0187 |
| hsa.miR.340.3p | -0.0172 |
| hsa.miR.187.3p | -0.0144 |
| hsa.miR.19a.3p | -0.0140 |
| hsa.miR.5699.5p | -0.0133 |
| hsa.miR.1.3p | -0.0101 |
| hsa.miR.6783.3p | -0.0048 |
| hsa.miR.129.5p | -0.0047 |
| hsa.miR.3150b.3p | -0.0045 |
| hsa.miR.4677.3p | -0.0032 |
| hsa.miR.219a.2.3p | -0.0029 |
| hsa.miR.6854.5p | -0.0012 |
| hsa.miR.144.5p | -0.0005 |
| hsa.miR.17.3p | -0.0002 |

**C**

Positive correlation with miR-155-5p

Negative correlation with miR-155-5p

| Genes | P-value | Pearson cor. coef. |
| --- | --- | --- |
| CCL5 | 6.59E-92 | 0.778 |
| NKG7 | 2.24E-86 | 0.763 |
| CD2 | 1.68E-85 | 0.76 |
| TRAC | 2.89E-84 | 0.756 |
| TRBC2 | 1.87E-83 | 0.754 |
| CD8A | 7.18E-83 | 0.752 |
| MIR155HG | 9.23E-83 | 0.752 |
| CD3D | 1.22E-82 | 0.752 |
| CD8B | 1.35E-82 | 0.752 |
| TIGIT | 3.33E-82 | 0.75 |
| UCK2 | 1.66E-15 | -0.364 |
| CASKIN1 | 1.45E-15 | -0.365 |
| SNHG29 | 1.02E-15 | -0.367 |
| PTK7 | 5.52E-16 | -0.37 |
| KCTD15 | 5.20E-16 | -0.37 |
| KIAA1549 | 3.07E-17 | -0.385 |
| GTF2IRD1 | 1.47E-17 | -0.388 |
| SOCS7 | 6.11E-18 | -0.392 |
| ADGRA3 | 1.88E-23 | -0.448 |
| AL031778.1 | 1.09E-23 | -0.45 |

**D**

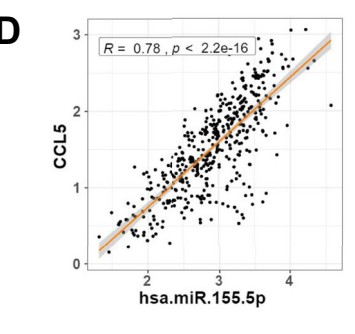

**E**

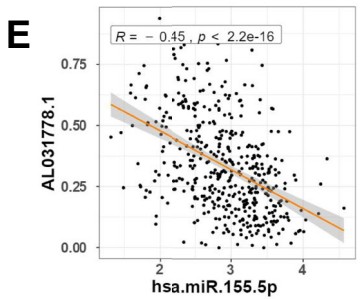

**F**

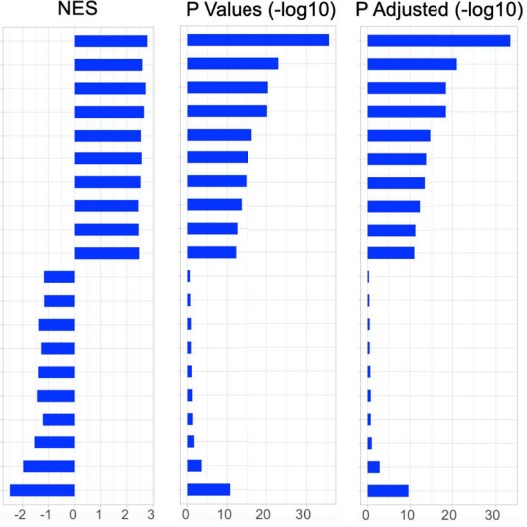

◀ **Figure EV2.  Lasso analysis of miRNAs against IFNγ signaling in melanoma (related to Fig. 4).**

(**A**) Mean-squared cross-validation (CV) error as a factor of increasing model penalty is shown ($n = 471$). (**B**) miRNA predictors with positive and negative coefficients at the lambda value corresponding to minimum CV error $+ 1$ standard error are shown. (**C**) Transcripts with the top positive and negative correlations with miR-155-5p are shown. (**D, E**) TCGEx scatter plot module was used to visualize top correlators of miR-155-5p. Statistical analysis was performed using Pearson correlation test, and the corresponding $P$ values are shown. (**F**) The GSEA results shown in Fig. 4E were exported from the TCGEx platform and plotted elsewhere to demonstrate that the numeric outputs can be reused by the users, if preferred.

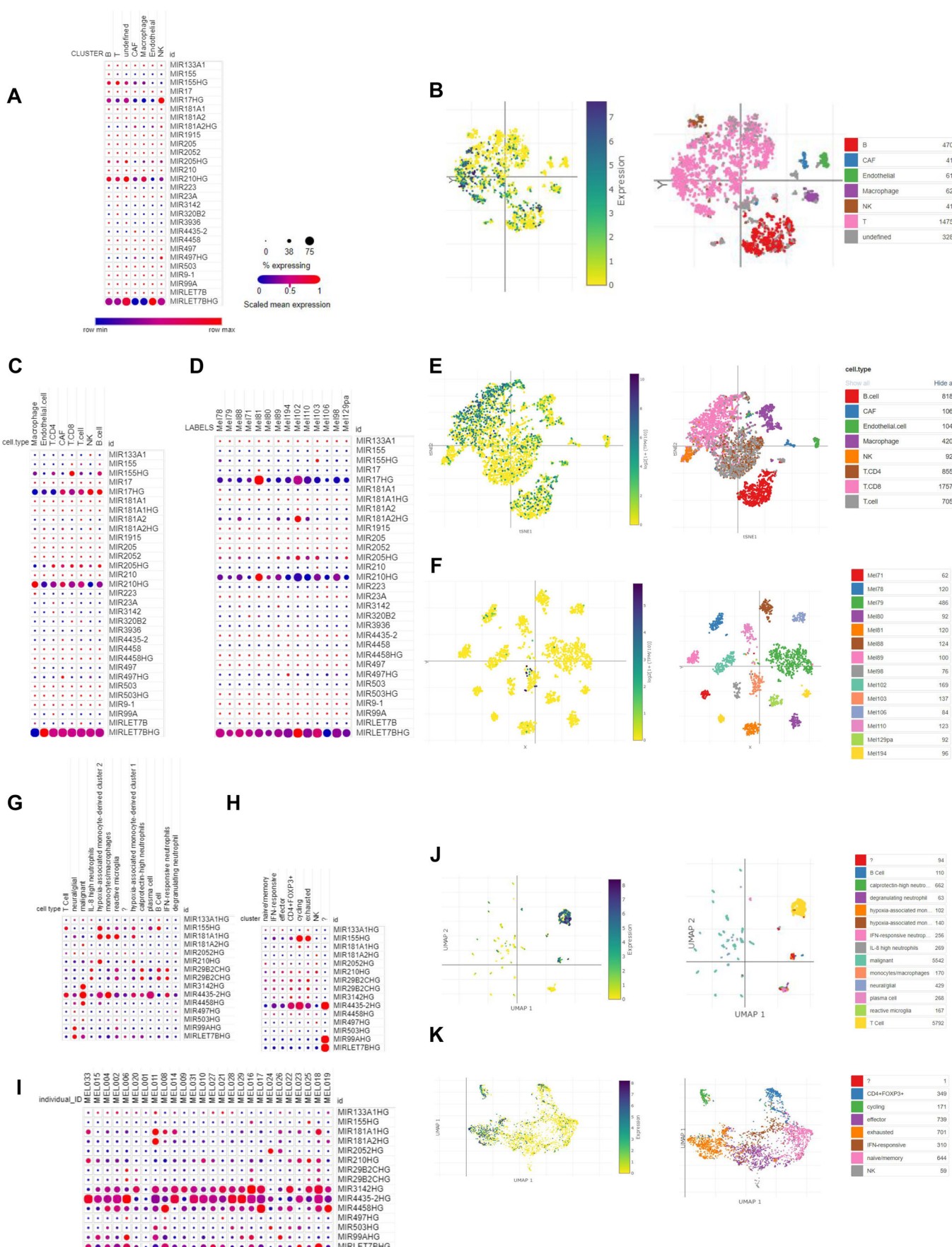

◀ **Figure EV3.  Publicly available scRNAseq data sets on Broad Single-Cell Portal were examined to investigate the expression patterns of ML-selected miRNAs.**

Three separate data sets were analyzed: SCP11 (**A, B**), SCP109 (**C–F**), and SCP1493 (**G–K**); and the Single-Cell Portal interface was used to generate the plots (query URLs can be found in the Appendix). (**A, C, D, G–I**). Dot plots indicate the expression of miRNAs in immune (**A, C, G, H**) and in melanoma (**D, G, I**) cell clusters. Some miRNAs show immune cell-selective expression patterns whereas others are more specific to melanoma cells within the TME. (**B, E, F, J, K**). UMAP plots showing the expression profile of MIR155HG in immune and tumor cell clusters reveals the T and B cell-selective expression of MIR155HG within the TME (**E, J**). MIR155HG was particularly elevated in cycling and exhausted T-cell clusters (**K**). In contrast, melanoma cells did not broadly express MIR155HG (**F, J**) across three studies.

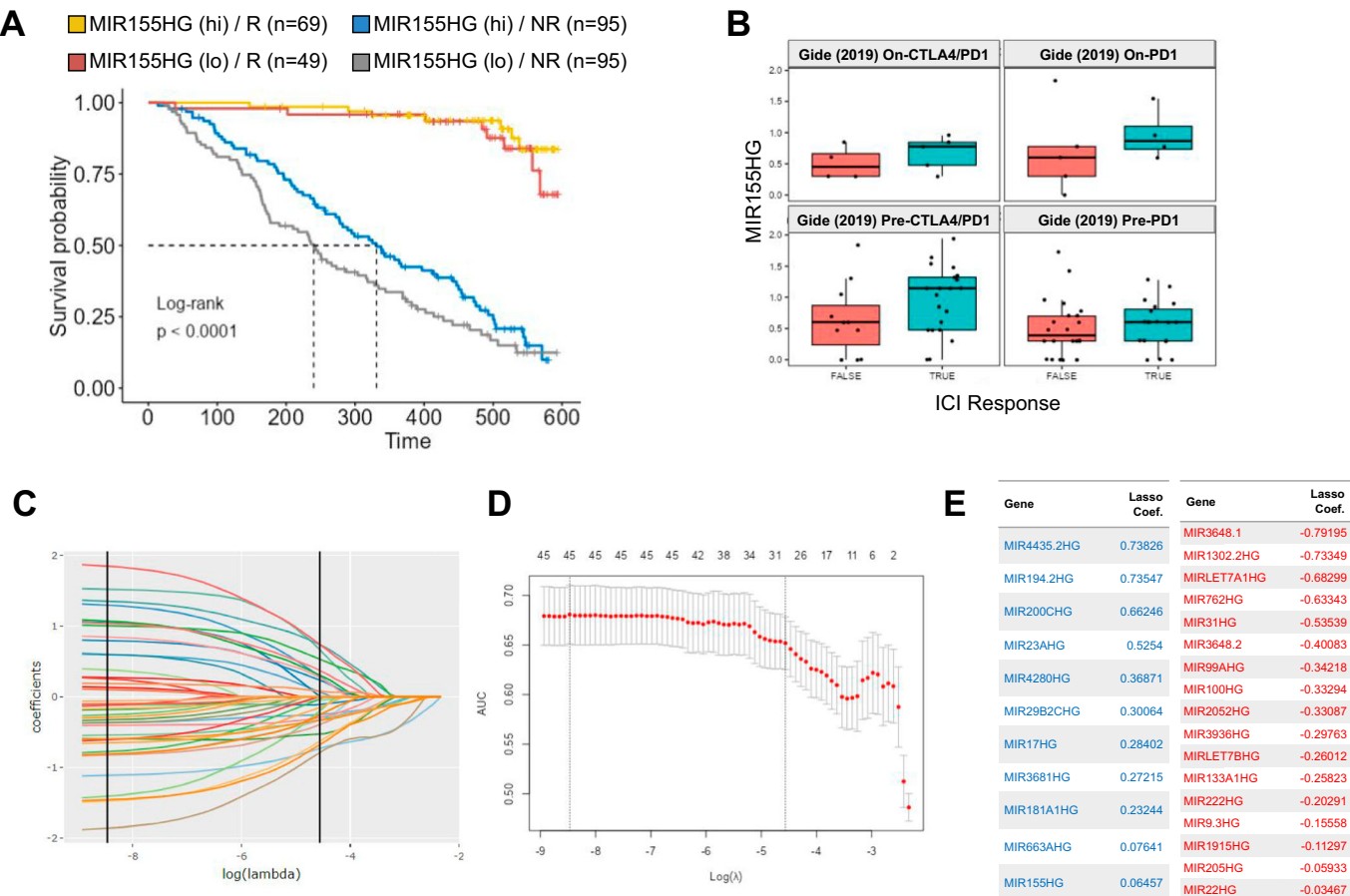

**Figure EV4.  Investigating the relationship between immunotherapy response and MIR155HG and other miRNA host genes (related to Fig. 6).**

(A) Kaplan–Meier survival analysis demonstrates that higher MIR155HG levels (categorized at median) prior to treatment correspond to improved survival in melanoma patients ($n = 313$); however, this trend was more emphasized in non-responders. (B) Boxplots show the MIR155HG transcript levels in responders and non-responders during or prior to ICI treatment (On-CTLA4/PD1 $n = 9$, Pre-CTLA4/PD1 $n = 32$, On-PD1 $n = 9$, Pre-PD1 $n = 41$). The box represents the interquartile range (IQR), spanning from the 25th percentile (lower bound) to the 75th percentile (upper bound). The horizontal line within the box indicates the median (50th percentile). Whiskers extend to the smallest and largest values within 1.5× IQR from the lower (Q1) and upper (Q3) quartiles, respectively. (C–E) Binomial lasso was performed using binarized response (CR/PR vs PD) as response variable and 45 miRNA host genes as predictors after low-expression filtering. Regularized coefficients (C), the cross-validated AUC of the penalized model (D), and nonzero coefficients at the minimum value of lambda within 1 standard error of the highest AUC (E) are shown in pre-treatment melanoma patients ($n = 313$). Error bars indicate the lambda values within one standard error of the highest cross-validated AUC.

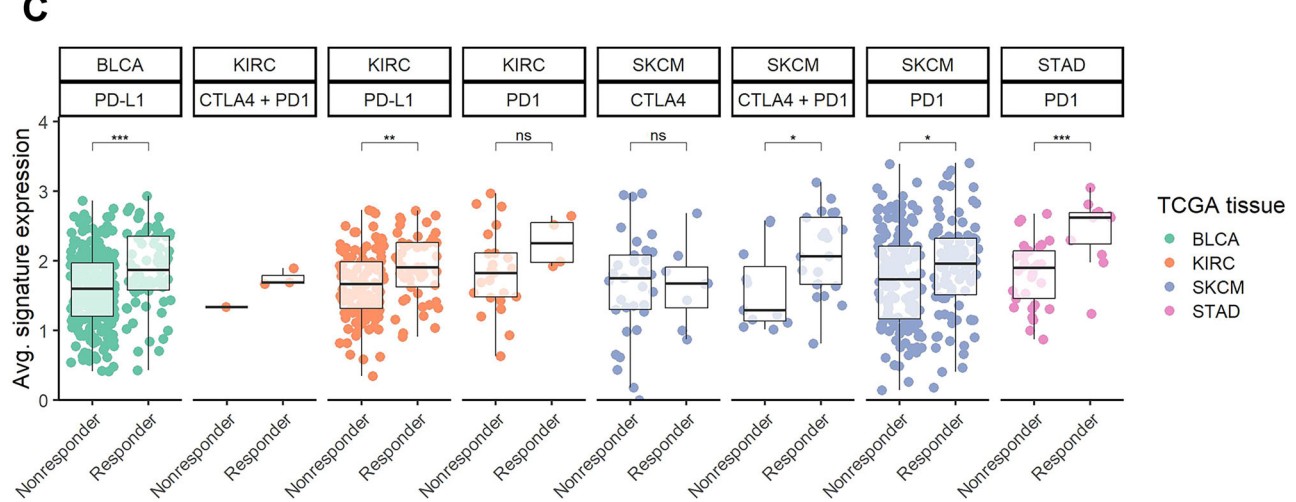

**Figure EV5. Examination of the Cytokine-5 signature as a new marker for immunotherapy response (related to Fig. 7).**

(A) The pairwise correlations between various signatures predicting immunotherapy response are depicted in a heatmap (figure was generated using the IOSig platform). The Pearson correlation coefficients were calculated by using the AUROC values across the studies in the IOSig database. (B) Pairwise correlations between the Cytokine-5 and IFNG6, IFN18 and Chemokine signatures are plotted in IOSig data sets. The best-fitting line (black) and its equation is shown along with the y = x diagonal line (red) indicating equality. Figure was generated in the R environment using the analysis results exported from the IOSig server. Statistical analysis was performed using Pearson correlation test, and the corresponding $P$ values are shown. (C) The boxplots illustrate the average Cytokine-5 signature expression in immunotherapy responders and non-responders across different cancer types in TCGEx data sets (BLCA/PD-L1 $n = 348$, KIRC/CTLA4 + PD1 $n = 4$, KIRC/PD-L1 $n = 176$, KIRC/PD1 $n = 27$, SKCM/CTLA4 $n = 42$, SKCM/CTLA4 + PD1 $n = 32$, SKCM/PD1 $n = 239$, STAD $n = 45$). The box represents the interquartile range (IQR), spanning from the 25th percentile (lower bound) to the 75th percentile (upper bound). The horizontal line within the box indicates the median (50th percentile). Whiskers extend to the smallest and largest values within 1.5× IQR from the lower (Q1) and upper (Q3) quartiles, respectively. $T$ test $P$ values are shown ($P < 0.001$, "***"; $P < 0.01$, "**"; $P < 0.05$, "*"; $P > 0.05$, "ns")

