## [Peer Review File · EMBO Reports]

TCGEx: A Powerful Visual Interface for Exploring and Analyzing Cancer Gene Expression Data

Muhammet Kus, Cagatay Sahin, Emre Kilic, Arda Askin, Mustafa Ozgur, Gokhan Karahanogullari, Ahmet Aksit, Ryan O'Connell, and Huseyin Ekiz

Corresponding author(s): Huseyin Ekiz (atakanekiz@iyte.edu.tr)

Review Timeline:

Submission Date:	29th Aug 24
Editorial Decision:	4th Nov 24
Revision Received:	13th Jan 25
Editorial Decision:	31st Jan 25
Revision Received:	12th Feb 25
Accepted:	17th Feb 25

Editor: Esther Schnapp

Transaction Report:

Dear Dr. Ekiz,

Thank you for your patience while your manuscript was peer-reviewed at EMBO reports. We have now received the full set of referee reports that is pasted below.

As you will see, the referees acknowledge that the TCGEx platform is potentially interesting. However, they also have several comments and suggestions for how the platform and the ms could be improved, and I think all suggestions should be addressed, especially the ones from referee 2. It is important that you compare TCGEx to Cbioportal in greater detail. Please let me know in case you have any comments or questions regarding the revisions and we can discuss this further, also in a video chat if you like.

I would thus like to invite you to revise your manuscript with the understanding that the referee concerns must be fully addressed and their suggestions taken on board. Please address all referee concerns in a complete point-by-point response. Acceptance of the manuscript will depend on a positive outcome of a second round of review. It is EMBO reports policy to allow a single round of major revision only and acceptance or rejection of the manuscript will therefore depend on the completeness of your responses included in the next, final version of the manuscript.

We realize that it is difficult to revise to a specific deadline. In the interest of protecting the conceptual advance provided by the work, we recommend a revision within 3 months (4th Feb 2025). Please discuss the revision progress ahead of this time with the editor if you require more time to complete the revisions.

- 1) A data availability section providing access to data deposited in public databases is missing. If you have not deposited any data, please add a sentence to the data availability section that explains that.
- 2) Your manuscript contains statistics and error bars based on $n=2$. Please use scatter blots in these cases. No statistics should be calculated if $n=2$.

3) We replaced Supplementary Information with Expanded View (EV) Figures and Tables that are collapsible/expandable online. A maximum of 5 EV Figures can be typeset. EV Figures should be cited as 'Figure EV1, Figure EV2' etc... in the text and their respective legends should be included in the main text after the legends of regular figures.

5) a complete author checklist, which you can download from our author guidelines <https://www.embopress.org/page/journal/14693178/authorguide>. Please insert information in the checklist that is also reflected in the manuscript. The completed author checklist will also be part of the RPF.

6) Please note that all corresponding authors are required to supply an ORCID ID for their name upon submission of a revised manuscript (<https://orcid.org/>). Please find instructions on how to link your ORCID ID to your account in our manuscript tracking system in our Author guidelines <https://www.embopress.org/page/journal/14693178/authorguide#authorshipguidelines>

12) All Materials and Methods need to be described in the main text using our 'Structured Methods' format, which is required for all research articles. According to this format, the Methods section includes a separate file called Reagents and Tools Table (listing key reagents, experimental models, software and relevant equipment and including their sources and relevant identifiers) and a Methods and Protocols section describing the methods using a step-by-step protocol format. The aim is to facilitate adoption of the methodologies across labs. More information on how to adhere to this format as well as a downloadable template (.docx) for the Reagents and Tools Table can be found in our author guidelines: <https://www.embopress.org/page/journal/14693178/authorguide#structuredmethods>.

An example of a Method paper with Structured Methods can be found here: <https://www.embopress.org/doi/full/10.1038/s44320-024-00037-6#sec-4>

I look forward to seeing a revised form of your manuscript when it is ready.

Referee #1:

In this study, the authors developed TCGEx web server which provide various bioinformatic analysis tools to handle RNA-seq and miRNA-seq data for multiple types of cancers profiled in the TCGA consortium. The authors applied TCGEx web server to melanoma data sets to investigate genotypic subgroups, immunological subgroups, and the factors associated with immunotherapy responses.

Comment1. In the Materials and Methods section in the page 4, line 78, the authors mentioned that RNAseq and miRNAseq data of TCGA were pre-processed for the web-based pipeline. What about other omics types such as proteome or genome (WGS) data. Are they available through the TCGEx tool?

Comment2. On page 5, line 102, the authors stated that they removed genes that are not expressed in 25% or more of the samples in each TCGA project. However, the criterion for an "expressed gene" needs to be clarified. Does it indicate genes with at least one read count, or does it refer to genes with expression levels above a certain cutoff?

Comment3. In the method section, TCGEx is well described. However, in the Result section, it is unclear which part is done using TCGEx, and which were not. Furthermore, the development of TCGEx should also be briefly described at the beginning of the Result section.

Comment4. On page12, line 334, it is reported that 'As expected from prior work, these miRNAs showed a strong positive correlation with T cell effector molecules including ...'. If they are target genes of those miRNAs, their expression would be expected to have a negative correlation with the expression of miRNAs. In this case, what would be the mechanism (or explanation) that leads to positive correlation between those miRNAs and T cell effector molecules? This needs to be described in detail.

Comment5. In Figure 4c-d, ROC curve shows the predictive power of 5 miRNAs compared to Hallmark IFN-gamma. What if the authors combine these features altogether? Will it predict the immune-enriched SKCM subsets better than each feature separately?

Comment6. In Figure2E, the unit of x-axis (time) is unclear. It would be helpful if the authors mention the unit (day, hour, minute...etc).

Comment7. In Figure 2A, the melanoma samples include both metastatic and primary solid tumor. It needs to be clarified whether those two classes show similar genetic/molecular phenotypes and survival curves in Figure 2C-E.

Comment8. In the Figure 3d-e, the authors analyzed the correlation between TCR diversity and intratumoral CD8+ T cell signatures within the TME. However, in the manuscript, it is unclear how they calculated TCR diversity. Is this TCR diversity derived from TCR-sequencing data, or inferred from the RNA-seq data of TCGA?

Comment9. In the Figure 4B, the legend is missing for the color bar. Additionally, the specific type of correlation coefficient (pearson or spearman) needs to be clarified.

Comment10. In the Figure 4E, the authors showed the actual values of NES, pval, and padj for different biological processes in the tables. Perhaps, it would be easier to understand if the authors visualized these values in a graphical format (bar graph, heatmap etc.)

Comment11. On page 13 (Figure 5), the authors investigated the predictive power of miRNA host gene expression in relation to cancer immunotherapy responses. As the authors mentioned, the RNA-seq data is based on the bulk tumor tissue samples,

meaning it includes a mixture of cancer cells and tumor microenvironment cells. In case the authors hypothesize that the miRNAs are acting within the TME and immunotherapy responses, it would be worth checking publicly available single-cell RNA-seq data, at least to validate that those miRNA host genes are expressed not only in the tumor cells but in the TME of melanoma samples.

Comment12. The link to the TCGEx webserver is not available.
<https://www.tcgex.iyte.edu.tr/>

Comment13. In Figure 6F-G, the analysis was performed for different cancer types. Are there particular cancer types that are more predictable in terms of immunotherapy response based on miRNA host genes compared to other cancer types?

Comment14. In the first part, the genetic subgroups of melanoma were investigated in detail. In the last part, the immunotherapy response of melanoma was shown in depth. How about the association of genotypes and immunotherapy response? Are there any association between these factors?

Referee #2:

This manuscript presents a web tool -tcgex- for the analysis of cancer transcriptome data. It combines miRNA/gene expression matrices and clinical variables from about 50 cohorts, with a particular focus on immune checkpoint inhibition (ICI) studies. Users can perform typical gene expression analyzes such as differential expression, correlation, survival, feature selection & classifiers. The authors then present 2 applications: 1) identification of miRNAs involved in ICI response and 2) development of a new ICI response signature based on cytokines.

I played with the tool and found it overall practical, responsive and working "as advertised", with a nice online help. However I have a few major comments about how the tool is presented and about the second use case.

1. I found the comparison with competing tools, especially Cbioportal, to be unsatisfying. The authors claim other tools lack capacity for "examining transcriptomics data from various sources including TCGA, landmark immunotherapy studies, and one's own research." It is true the main Cbioportal server does not have much immunotherapy studies, however it has a worldwide recognition and can be installed locally to analyze users' own data, and it includes modules for survival analysis, volcano plots, coexpression, oncoplots etc. I think the authors should list exactly what functions are possible with tcgex that are not possible Cbioportal or other servers such as iosig.

2. The manuscript lacks detail on how the clinical and gene expression variables were integrated.

- For instance in original clinical datasets, response can be encoded under different variable names (responder, response, RECIST_best_response, etc). How is it curated in the database in order to enable multi cohorts analyses? It seems to me that there is no uniform response variable that can be used pan-cohort. What about other clinical variables?

- For gene expression counts: CPM, TPM, RPKM cannot be combined in the same analysis. The authors acknowledge this, but they should also provide recommendations on how to deal with this. Also, does the interface warn users when attempting to combine incompatible counts? Anyway, even the same count type (eg TPM) can be quite variable depending on the software and reference transcriptome used. This should be acknowledged.

3. I found the focus of the paper a bit too meandering. The story goes from (A) presenting the tool, to (B) showcasing an application to miRNA discovery in relation to ICI response, to (C) discovery of a new pan-cancer ICI response signatures based on cytokines. This may be too much for a single paper; See next point.

4. In part (C) above, the authors claim that a new signature based on 5 cytokines performs better in pan-cancer response precision than any prior published signature. This is an important claim that warrants a reproducible computational workflow. The problem is that the author's methods only involve point and click on visual servers. Thus the claim is hard to reproduce. In my opinion this part should be the object of an independent publication. The miR-155 application does not make such strong claim and, in my opinion, would be sufficient to showcase the tool.

Other comments

L 425-455: The Cytokine-5 signature is discovered using a pan cancer subset of TCGex, then validated on another dataset on the IOSig platform. However it is not clear whether the two datasets overlap or not. They should not.

L. 454-456. My understanding was that the Cytokine-5 signature was discovered on the TCGEx ICI studies, thus I do not understand the value of finding it higher in responders in the same data. Or perhaps it is not the same data, but this needs to be clarified.

"We also noted that the expression of this signature increased during ICI treatment regardless of the response classification

suggesting that the Cytokine-5 signature is induced upon immunotherapy, albeit more strongly in responding tumors."
-> is this result shown?

I would not use IMPRES for comparison with the Cytokine-5 signature, as it has been strongly criticized
<https://pubmed.ncbi.nlm.nih.gov/31806907/>

L. 120. "gene expression can be categorized across all the projects at a common threshold, "
Let's say I select 3 datasets with different count methods and ask for a Cox analysis. How can a single threshold work for all 3 datasets?

L. 370 "by integrating 10 quartile-normalized ICI studies "
-> Please develop. Is this normalization method available through the visual interface? How?

Line 334: "As expected from prior work," -> reference needed

The protocol for reproducing mir-155 results in Fig. 4,5 should be available, even if it was performed through the visual interface

Could the author say a word about future update policy?

I found the main figures were sometimes too crowded, with detailed statistics (Lasso coefficients, GSEA gene rank plots) that could easily go to supplementary figures.

About the visual interface:

- In the heatmap view, it would be useful to allow grouping of samples by categorical variables. Unless I missed this option.
- Can one select more than 1 gene for boxplots?
- It seems that gene expression plots have no unit on the Y axis
- Total sample counts in selected datasets are not readily visible
- There are many parasite features to chose from that mask the interesting ones. For instance to plot gene sets against response, one has to go through the whole list of human genes which are considered there as features.

Referee #3:

This manuscript introduces The Cancer Genome Explorer (TCGEx), a novel web-based interface designed for integrated cancer transcriptomics analysis, specifically aiming to facilitate non-specialists' access to large datasets like those from The Cancer Genome Atlas (TCGA). TCGEx provides pre-processed transcriptomics data and supports data from immune checkpoint inhibition studies, allowing researchers to upload their own datasets for analysis. Built on the R/Shiny framework, TCGEx includes ten modules for various types of analysis, such as survival modeling, gene set enrichment analysis (GSEA), principal component analysis (PCA), and machine learning for feature selection and prediction.

Using this platform, the authors identified immune markers and microRNAs associated with immunotherapy response in melanoma, highlighting a cytokine signature predictive of positive responses to checkpoint inhibitors across multiple cancers. The authors validated these findings in independent datasets. The web application aims to democratize high-throughput cancer data analysis, making it more accessible to researchers across disciplines by providing user-friendly, customizable analysis tools and publication-ready visual outputs.

TCGEx platform has many strengths. It provides a user-friendly interface for broad accessibility and several comprehensive analysis modules. The platform allows researchers to integrate TCGEx into their workflows or adapt the code for specific needs. This open-access approach promotes transparency and enhances reproducibility. Finally the platform provides publication-ready plots.

The only minor weakness is that the authors use gender when they mean sex both in the manuscript and on their website. As far as I know the underlying TCGA dataset does not have gender information. It has the sex of the donors. This should be corrected in both the manuscript and on the website.

Dear Editor,

We thank the reviewers for their valuable insights and we are glad to see that they found our newly developed program “practical”, “responsive”, and “having many strengths”. Below, we provide a point-by-point response addressing the reviewers' comments in entirety, which significantly improved the TCGEx platform and its manuscript. Revised text is highlighted in blue and comments are added to ease tracking of specific responses (eg. R1C4 corresponds to the 1st reviewer's 4th point). Please let us know if anything else is needed.

We look forward to your favorable assessment.

Best regards,
Atakan

Ref 1

In this study, the authors developed TCGEx web server which provide various bioinformatic analysis tools to handle RNA-seq and miRNA-seq data for multiple types of cancers profiled in the TCGA consortium. The authors applied TCGEx web server to melanoma data sets to investigate genotypic subgroups, immunological subgroups, and the factors associated with immunotherapy responses.

Comment1. In the Materials and Methods section in the page 4, line 78, the authors mentioned that RNAseq and miRNAseq data of TCGA were pre-processed for the web-based pipeline. What about other omics types such as proteome or genome (WGS) data. Are they available through the TCGEx tool?

We initially developed TCGEx with transcriptomics data in mind, however, the visualization and analysis modules can also work with other types of gene expression data including proteomics data. Since TCGEx is focused on functional genomics analyses, it was not designed to examine genomic variations in WGS data. Based on this valuable comment, we now included the reverse phase protein array (RPPA) proteomics data in the preprocessed TCGA datasets. These features are accessible with the prefix “prt” (e.g. prt.Akt2) and can be utilized in the analysis pipelines. To demonstrate this data type, we plotted RNA and protein levels of a pan leukocyte marker, PTPRC (CD45), and showed improved survival outcomes in tumors with higher levels of this protein and its transcript (Fig.S3).

Comment2. On page 5, line 102, the authors stated that they removed genes that are not expressed in 25% or more of the samples in each TCGA project. However, the criterion for an “expressed gene” needs to be clarified. Does it indicate genes with at least one read count, or does it refer to genes with expression levels above a certain cutoff?

We preprocessed RNAseq and miRNAseq data by $\log_{10}(\text{cpm}+1)$ transformation. We removed genes where the expression was zero or not-available (NA, as in the case for some samples due to lack of miRNAseq data) for 25% or more of the samples in the data set. With this approach, we aimed to minimize outlier effects in the analyses and reduce the data size for improved responsiveness. We added further clarifications in the manuscript text.

Comment3. In the method section, TCGEx is well described. However, in the Result section, it is unclear which part is done using TCGEx, and which were not. Furthermore, the development of TCGEx should also be briefly described at the beginning of the Result section.

All figures in the manuscript were obtained using the TCGEx interface except for the IOSig analyses (**Fig 6F, Fig7D-F**). This part is clarified and a brief description is added at the beginning of the Results section as suggested.

Comment4. On page12, line 334, it is reported that 'As expected from prior work, these miRNAs showed a strong positive correlation with T cell effector molecules including ...'. If they are target genes of those miRNAs, their expression would be expected to have a negative correlation with the expression of miRNAs. In this case, what would be the mechanism (or explanation) that leads to positive correlation between those miRNAs and T cell effector molecules? This needs to be described in detail.

We thank the reviewer for highlighting this important point. We realize this section may have sounded as if these protein-coding genes are direct targets of miRNAs, but that was not the case as indicated in the TargetScan database. Rather, these miRNAs were found to be associated with T cell effector molecules in various studies in the field. Our goal was to emphasize that the TCGEx findings are further supported by the literature and we added references accordingly.

However, the raised point is valuable and warrants a short discussion. Observing positive correlations between miRNAs and their target mRNAs is possible due to a number of reasons. First, miRNAs can have hundreds of target mRNAs and their expression does not always negatively correlate with all of its targets. Second, the bulk expression data may not provide the high resolution which enables studying miRNA-mRNA interactions at the cellular level. In this case, while the miRNA expression in a cell can suppress its specific target mRNA, the same mRNA can be expressed at high levels in other cells within the tumor microenvironment. Thus the bulk snapshot may reveal a positive correlation between miRNA and mRNA. Third, an imperfect miRNA-mRNA base pairing can lead to translational suppression in the absence of transcript degradation. This comment is now addressed at.

Comment5. In Figure 4c-d, ROC curve shows the predictive power of 5 miRNAs compared to Hallmark IFN-gamma. What if the authors combine these features altogether? Will it predict the immune-enriched SKCM subsets better than each feature separately?

We performed analyses to address this interesting thought. Combination of these features resulted in an AUC value of 0.893, while the AUC of 5-miRNA signature was 0.913. Thus, adding the Hallmark IFN γ genes to the selected miRNAs did not improve the prediction of the immune-enriched subset in melanoma. On a related point, we noticed a typo in the AUC annotations on the graph at the initial submission and we corrected these values in the revision (miR-155 AUC:0.826; IFN γ AUC: 0.887; 5-miRNA AUC: 0.913). This finding is added to the manuscript.

Comment6. In Figure2E, the unit of x-axis (time) is unclear. It would be helpful if the authors mention the unit (day, hour, minute...etc).

Since the TCGEx can work with user-provided data sets with different time scales, we chose “Time” as the axis title for general use. However, the x-axis unit is clarified as “Days” in the manuscript.

Comment7. In Figure 2A, the melanoma samples include both metastatic and primary solid tumor. It needs to be clarified whether those two classes show similar genetic/molecular phenotypes and survival curves in Figure 2C-E.

We added new supplementary figures to address this comment (**Fig.S1a, S2c-f**). Briefly, the selected markers showed mostly similar expression profiles between mutational subtypes in metastatic and primary tumors, although the latter group contained fewer samples. Possibly due to low sample numbers, survival differences were not readily observable in primary tumor molecular subsets.

Comment8. In the Figure 3d-e, the authors analyzed the correlation between TCR diversity and intratumoral CD8+ T cell signatures within the TME. However, in the manuscript, it is unclear how they calculated TCR diversity. Is this TCR diversity derived from TCR-sequencing data, or inferred from the RNA-seq data of TCGA?

We thank the reviewer for the opportunity to clarify this point. We did not perform specific analyses to quantify the TCR diversity. Rather, we utilized previously published data from the Thorsson et al. study (Cell, 2018). As described in this manuscript, TCR diversity is inferred from the tumor RNAseq data through the identification of polymorphic CDR3 sequences. Antigen receptor diversity data were integrated into our datasets to facilitate analyses exemplified in Fig3d-e.

Comment9. In the Figure 4B, the legend is missing for the color bar. Additionally, the specific type of correlation coefficient (pearson or spearman) needs to be clarified.

We updated the figure legend to clarify that Spearman’s method is used in the analysis.

Comment10. In the Figure 4E, the authors showed the actual values of NES, pval, and padj for different biological processes in the tables. Perhaps, it would be easier to understand if the authors visualized these values in a graphical format (bar graph, heatmap etc.)

We agree with the reviewer that a graphical format may be easier for understanding the GSEA results. The reason we wanted to show the actual values of NES, pval, padj and the ranked gene list (black barcodes in the middle) was emphasizing the two ends of the enrichment results (ie. enrichment in the “high” and “low” categories). This graph is conveniently generated in the TCGEx GSEA module and it enables a quick exploration of the most differentially enriched pathways. Thus, we wanted to demonstrate what users would see on the platform. Still, to address this point, we generated a supplementary figure from the results (**Fig.S4f**) and described that numerical GSEA output can be exported to generate plots elsewhere.

Comment11. On page 13 (Figure 5), the authors investigated the predictive power of miRNA host gene expression in relation to cancer immunotherapy responses. As the authors mentioned, the RNA-seq data is based on the bulk tumor tissue samples, meaning it includes a mixture of cancer cells and tumor microenvironment cells. In case the authors hypothesize that the miRNAs are acting within the TME and immunotherapy responses, it would be worth checking publicly available single-cell RNA-seq data, at least to validate that those miRNA host genes are expressed not only in the tumor cells but in the TME of melanoma samples.

We thank the reviewer for this interesting suggestion. Indeed, miRNAs can be expressed both in tumor cells and infiltrating immune cells to control tumor progression and immunity. We examined three publicly available melanoma scRNAseq datasets to reveal expression patterns of selected miRNAs at the single cell resolution. Our findings are shown in a new supplementary figure (**Fig.S6**) and described in the manuscript. Briefly, while miRNA expression patterns were variable across cell types and studies, we consistently identified MIR155HG as a T cell-selective miRNA within the melanoma TME suggesting this miRNA and others can function in the infiltrating immune cells.

Comment12. The link to the TCGEx webserver is not available.

We apologize for the typo here. The URL is updated as <https://tcgex.iyte.edu.tr/>.

Comment13. In Figure 6F-G, the analysis was performed for different cancer types. Are there particular cancer types that are more predictable in terms of immunotherapy response based on miRNA host genes compared to other cancer types?

We added a new supplementary figure to examine MIR155HG AUC values in different tumor types (**Fig.S8e**). Due to variation across data datasets and small sample sizes it is not straightforward to identify tumor types where MIR155HG expression consistently predicted better responses.

Comment14. In the first part, the genetic subgroups of melanoma were investigated in detail. In the last part, the immunotherapy response of melanoma was shown in depth. How about the association of genotypes and immunotherapy response? Are there any association between these factors?

Melanoma mutational subtypes were shown to have differential survival outcomes and immunotherapy responses (doi:10.1158/1078-0432.CCR-09-2509) (doi:10.3390/cancers12092359). We performed chi-square analysis to examine the relationship between the mutational subtypes and the immunotherapy response in the CRI-iAtlas melanoma data sets featured on TCGEx. We found no statistically significant associations between therapy response (R/NR) and the mutational subsets (BRAF, NRAS, NF1, WT) ($p=0.2021$), however the low number of patients in each group could pose a challenge to identify true relationships. To approach this question visually, we also plotted the immune-associated gene expression between responders and non-responders in different mutational subsets and included this as a supplementary figure.

...

Ref 2

This manuscript presents a web tool -tcgex- for the analysis of cancer transcriptome data. It combines miRNA/gene expression matrices and clinical variables from about 50 cohorts, with a particular focus on immune checkpoint inhibition (ICI) studies. Users can perform typical gene expression analyzes such as differential expression, correlation, survival, feature selection & classifiers. The authors then present 2 applications: 1) identification of miRNAs involved in ICI response and 2) development of a new ICI response signature based on cytokines.

I played with the tool and found it overall practical, responsive and working "as advertised", with a nice online help. However I have a few major comments about how the tool is presented and about the second use case.

1- I found the comparison with competing tools, especially Cbioportal, to be unsatisfying. The authors claim other tools lack capacity for "examining transcriptomics data from various sources including TCGA, landmark immunotherapy studies, and one's own research." It is true the main Cbioportal server does not have much immunotherapy studies, however it has a worldwide recognition and can be installed locally to analyze users' own data, and it includes modules for survival analysis, volcano plots, coexpression, oncoplots etc. I think the authors should list exactly what functions are possible with tcgex that are not possible with Cbioportal or other servers such as IOsig.

We appreciate the reviewer's positive feedback and welcome the opportunity to further highlight the distinctive features of TCGEx. We agree with the reviewer that cBioportal is widely adopted in the field and it includes some of the capabilities of TCGEx. We added a supplementary table comparing TCGEx with cBioportal and several other web-based analysis interfaces (**Table.S7**). Briefly, while cBioportal particularly facilitates exploring genomic variants in cancer, TCGEx aims to provide access to functional genomics analysis algorithms. Although both tools can assist with survival analysis and examination of gene expression data against clinical features, TCGEx offers more flexibility and easier overall user experience (more below). For instance, in survival modeling, TCGEx allows multivariable analyses as well as user-friendly options to redefine sample subsets and expression cut-offs. Similarly, in scatter plot and boxplot modules, TCGEx allows faceting the graphs by clinical meta features as exemplified in the manuscript. Additionally, TCGEx enables users to utilize unsupervised and supervised machine learning algorithms and perform gene set enrichment analyses in specific data subsets, which are not possible in cBioportal. Taken together, the features summarized here distinguish TCGEx from other tools in the field. As part of the revision, we also included the reverse phase protein array (RPPA) proteomics data from the TCGA studies which further expanded the uses of the platform.

While the user experience can be subjective and hard to quantify, we attempted a head-to-head comparison between TCGEx and cBioportal for three overlapping features: Kaplan-Meier survival modeling, gene-to-gene correlation analysis, gene-to-category plotting. Using the same computer system (64-bit Win11, i7 2.4 ghz, 16 gb RAM, Chrome browser), we performed analyses on both TCGEx and cBioportal, and captured the computer screen to analyze our interactions with these tool (the recorded screencast videos can be found in the shared GoogleDrive folder: <https://drive.google.com/drive/folders/1EBdphcySADRAOcQFGXqou0pdsDLMewDO?usp=sharing>) (we will also make these videos available on a dedicated YouTube Channel later). As seen in the table below, both platforms required a similar number of clicks and comparable time to complete the tasks for simple analyses, although TCGEx was slightly faster. However, more customized analyses (such as plotting gene expression between males and females in different mutational subtypes, survival analysis with miRNAseq/proteomics data) would require several clicks in cBioportal or would be not possible at all, unlike TCGEx. We could not perform a head-

to-head comparison for the following TCGEx capabilities because they do not currently have a counterpart in cBioportal: gene-to-gene correlation matrix, gene set enrichment, heatmap/hierarchical clustering, principal component analysis, lasso/ridge/elastic net machine learning, multivariable cox proportional hazards analysis, examination of miRNAseq and proteomics data. Notably, cBioportal includes data types and analyses that are not available in TCGEx. Thus we do not envision TCGEx to replace cBioportal; instead it arises as a strong tool complementing the existing tools. In summary, we believe that TCGEx will reduce the barriers to high throughput data and enable researchers from various backgrounds to benefit from public and private datasets.

Comparison of TCGEx and cBioPortal Analyses

Analysis Type	TCGEx	TCGEx	cBioPortal	cBioPortal
	Click Count	Elapsed Time (sec)	Click Count	Elapsed Time (sec)
Kaplan-Meier Survival Analysis	10	21	12	32
Gene vs Gene Scatter Plot	12	25	17	27
Gene vs Category Plots	11	21	14	27

2- The manuscript lacks detail on how the clinical and gene expression variables were integrated.

- For instance in original clinical datasets, response can be encoded under different variable names (responder, response, RECIST_best_response, etc). How is it curated in the database in order to enable multi cohorts analyses? It seems to me that there is no uniform response variable that can be used pan-cohort. What about other clinical variables?

We provided further clarifications on how the data were integrated. Briefly, we used the metadata variable names directly from the repositories we pull data from (CRI-iAtlas, cBioportal) without alterations. Variables “meta.response” and “meta.responder”, for instance, are found in the quartile-normalized CRI-iAtlas datasets and they describe the type (CR, PR, PD) and classification (R, NR) of therapy response consistently across datasets, so they can be used in pan-cancer analyses. Data sets originating from the cBioportal repository have more diverse variable names and normalization methods making integration more challenging or not appropriate (see below for built-in warnings). In these data sets, we chose not to alter naming of the variables to be consistent with the original data and associated manuscripts. We just curated and harmonized a few generic variables (eg. meta.vital_status, meta.ICB, etc.) to be consistent with the working pipeline. The code to reproduce these datasets are shared in the TCGEx GitHub repository.

Upon selecting multiple compatible datasets, the TCGEx pipeline retains only the common variables for downstream analyses. For instance, when integrating TCGA melanoma and lung cancer data, “meta.sex” will be retained as it is a common variable; however, “meta.pigment.score” will be dropped as it is specific to melanoma samples only.

3- For gene expression counts: CPM, TPM, RPKM cannot be combined in the same analysis. The authors acknowledge this, but they should also provide recommendations on how to deal with this. Also, does the interface warn users when attempting to combine incompatible counts? Anyway, even the same count type (eg TPM) can be quite variable depending on the software and reference transcriptome used. This should be acknowledged.

We thank the reviewer for his/her insights and this excellent suggestion. We updated the code to prevent calculations and prompt a warning message when different normalization types are selected. We also updated the manuscript to acknowledge the possible shortcomings of integration.

4- I found the focus of the paper a bit too meandering. The story goes from (A) presenting the tool, to (B) showcasing an application to miRNA discovery in relation to ICI response, to (C) discovery of a new pan-cancer ICI response signatures based on cytokines. This may be too much for a single paper; See next point.

We appreciate the concern here about the focus of the paper. Our manuscript intends to present TCGEx as a tool that can be used in various research contexts. As such, we first described a use-case scenario based on our lab's ongoing work on miRNAs in the context of antitumor immunity. Although we identified interesting relationships between intratumoral miRNA expression and the IFNg response signatures, we didn't find a strong association with the ICI response. We then pivoted to cytokine/cytokine receptor genes, because they are more broadly expressed in tumors and they do not require special detection methods (such as short RNA sequencing) to be studied across several studies. With this approach, we identified a short list of genes that can distinguish ICI responders comparably to the existing signatures (please see our response to the next comment). Taken together, we demonstrated how users with different research hypotheses benefit from the TCGEx platform. Proper adjustments were made to the text to address this comment.

5- In part (C) above, the authors claim that a new signature based on 5 cytokines performs better in pan-cancer response prediction than any prior published signature. This is an important claim that warrants a reproducible computational workflow. The problem is that the author's methods only involve point and click on visual servers. Thus the claim is hard to reproduce. In my opinion this part should be the object of an independent publication. The miR-155 application does not make such a strong claim and, in my opinion, would be sufficient to showcase the tool.

We agree with the reviewer that our claim may be too strong without further validations, and we made necessary adjustments in the text. It is important to clarify that the primary purpose of this manuscript is not to assert a claim about the superiority of the Cytokine-5 signature; instead our goal was to present a scenario that demonstrates the capabilities of TCGEx in enabling hypothesis generation. Although the results are based on point-and-click visual servers, all the analyses are reproducible both locally and on the web using the parameters described in the manuscript. We believe it is fair to say that the discovery of the Cytokine-5 signature demonstrates the capability of TCGEx in identifying biomarkers that can be applicable to a wider context. Our findings indicate that the Cytokine-5 signature -which has partial overlaps with some of the other signatures- could perform comparably or as good as to previously defined signatures in multiple data sets. By showcasing how the platform can be used to interrogate various data sets, we aim to highlight its utility as a robust tool for research, and we will pursue further validation studies in the future.

Other comments

6- L 425-455: The Cytokine-5 signature is discovered using a pan cancer subset of TCGex, then validated on another dataset on the IOSig platform. However it is not clear whether the two datasets overlap or not. They should not.

We appreciate the reviewer's valuable insights here. In Fig.7d, the Cytokine-5 score is assessed in each of the IOSig datasets separately. The pan-cancer studies in TCGEx were included in this heatmap as reference and to validate their high AUCs in a different analysis platform, because it does not affect the rest of the comparisons. However, we updated Fig.7e where the mean AUC was shown across IOSig datasets by removing the TCGEx pan-cancer studies. The results did not change significantly compared to the first submission, in that, the Cytokine-5 score still had the highest average AUC across 23 datasets, although some of the other signatures changed position in the graph.

7- L. 454-456. My understanding was that the Cytokine-5 signature was discovered on the TCGEx ICI studies, thus I do not understand the value of finding it higher in responders in the same data. Or perhaps it is not the same data, but this needs to be clarified.

We agree with the reviewer's view that Cytokine-5 score is expected to be higher in responders in the TCGEx data sets. We included Fig7g to demonstrate the distribution and range of the data points in different data sets, and updated the manuscript as such. Such graphs can be easily generated using TCGEx's source data that are publicly available through Figshare and Dockerized TCGEx container.

8- "We also noted that the expression of this signature increased during ICI treatment regardless of the response classification suggesting that the Cytokine-5 signature is induced upon immunotherapy, albeit more strongly in responding tumors."

-> is this result shown?

Fig.7h demonstrates this assessment and we moved the figure citation to the end of this sentence. Briefly, in non-responder patients, the median of the Cytokine-5 genes' averaged expression increased from ~1.8 pre-treatment to ~2 during treatment. In responder patients, the average expression rose from approximately 2.1 pre-treatment to around 2.5 during treatment (quartile normalized count data). These results indicate that, irrespective of the response classification, the expression of the Cytokine-5 signature is upregulated with ICI treatment, with a more pronounced induction observed in responding tumors.

9- I would not use IMPRES for comparison with the Cytokine-5 signature, as it has been strongly criticized <https://pubmed.ncbi.nlm.nih.gov/31806907/>

We thank the reviewer for this insightful contribution. We are aware of the ongoing debate regarding the IMPRES signature, including the criticisms raised by the mentioned study from Carter et al. and the subsequent response from Auslander et al. addressing these concerns (Auslander et al., Nat Med 2019, doi:10.1038/s41591-019-0646-5). As both studies remain in circulation, we decided not to take a position in this debate, but we rather wanted to provide a comprehensive evaluation of the Cytokine-5 signature against a range of immune-related signatures that have been previously reported in the literature. As IMPRES is only one of these signatures and there are many others in our analyses, we would like to respectfully include it in the analyses. However, we acknowledged this point in the revised manuscript.

10- L. 120. "gene expression can be categorized across all the projects at a common threshold, " Let's say I select 3 datasets with different count methods and ask for a Cox analysis. How can a single threshold work for all 3 datasets?

We appreciate the reviewer's detailed analysis and now we updated TCGEx to prevent integration of datasets with different count methods. Previously, this would have been allowed and lead to erroneous results. As addressed in the Point-3, the manuscript is updated to inform the users about potential shortcomings of integrative analyses.

11- L. 370 "by integrating 10 quartile-normalized ICI studies "

-> Please develop. Is this normalization method available through the visual interface? How?

The quartile-normalized ICI data used in TCGEx is sourced directly from CRI, and this pre-normalized data set can be utilized within the TCGEx interface. We provide easy access to the entire data object (pan-cancer) or its subsets "sliced" based on the study. We added further details to the manuscript to better describe this data sets.

12- Line 334: "As expected from prior work," -> reference needed

References are added to this section.

13- The protocol for reproducing mir-155 results in Fig. 4,5 should be available, even if it was performed through the visual interface

We added step-by-step instructions to reproduce figure 4 and 5 in the supplement.

14- Could the author say a word about future update policy?

We will actively monitor GitHub repository issues to engage with the users and address potential bugs/problems and feature requests. As our time permits, we will incorporate new data sets into the platform and test the pipeline with different types of data including Nanostring, ChIP-seq, and metabolomics data. Although we think most of the modules should work with other data types, their accuracy and robustness remain to be tested. We will share updates on the TCGEx website as well as the GitHub repository. We also would like to create a YouTube channel to demonstrate the capabilities of TCGEx to new users to facilitate adoption.

15- I found the main figures were sometimes too crowded, with detailed statistics (Lasso coefficients, GSEA gene rank plots) that could easily go to supplementary figures.

We appreciate the view of the reviewer and agree that some figures may be rather crowded. We wanted to include these figure panels to demonstrate the output a user would get when using the TCGEx platform. We would like to still show these figure panels unless the journal guidelines dictate otherwise, because some readers may want to see the coefficients and GSEA plots in the main body of the manuscript and it may be easier to understand the analyses with visual cues.

16- About the visual interface:

- In the heatmap view, it would be useful to allow grouping of samples by categorical variables. Unless I missed this option.

The heatmap module already allows selecting categorical variables to show in an annotation bar. In this way, clinical metadata can be visualized along with gene expression.

- Can one select more than 1 gene for boxplots?

Selecting more than one gene for boxplots is not currently possible, but we are hoping to make this feature available in the future. At this time, correlation and heatmap modules can address the need to examine multiple genes at the same time.

- It seems that gene expression plots have no unit on the Y axis

We deliberately did not hard-code the unit on the Y axis, because TCGEx provides access to data sets with different normalization methods. These normalization methods are listed in the data selection module and they are visible to users as they start their analysis.

- Total sample counts in selected datasets are not readily visible

The total sample numbers can be seen upon hovering the cursor on the pie and/or the column charts in the data selection module.

- There are many parasite features to choose from that mask the interesting ones. For instance to plot gene sets against response, one has to go through the whole list of human genes which are considered there as features.

We understand how the drop-down menus may appear cluttered with uninteresting features at the first glance. These selection boxes were developed using the SelectizeInput Shiny option and they can do partial character mapping using regular expressions. Thus, when the user starts typing the feature of interest, the selection box suggests relevant variable names. This design circumvents the need for going through the list of numerous features in the datasets.

.....

Ref 3

This manuscript introduces The Cancer Genome Explorer (TCGEx), a novel web-based interface designed for integrated cancer transcriptomics analysis, specifically aiming to facilitate non-specialists' access to large datasets like those from The Cancer Genome Atlas (TCGA). TCGEx provides pre-processed transcriptomics data and supports data from immune checkpoint inhibition studies, allowing researchers to upload their own datasets for analysis. Built on the R/Shiny framework, TCGEx includes ten modules for various types of analysis, such as survival modeling, gene set enrichment analysis (GSEA), principal component analysis (PCA), and machine learning for feature selection and prediction.

Using this platform, the authors identified immune markers and microRNAs associated with immunotherapy response in melanoma, highlighting a cytokine signature predictive of positive responses to checkpoint inhibitors across multiple cancers. The authors validated these findings in independent datasets. The web application aims to democratize high-throughput cancer data analysis, making it more accessible to researchers across disciplines by providing user-friendly, customizable analysis tools and publication-ready visual outputs.

TCGEx platform has many strengths. It provides a user-friendly interface for broad accessibility and several comprehensive analysis modules. The platform allows researchers to integrate TCGEx into their workflows or adapt the code for specific needs. This open-access approach promotes transparency and enhances reproducibility. Finally the platform provides publication-ready plots.

The only minor weakness is that the authors use gender when they mean sex both in the manuscript and on their website. As far as I know the underlying TCGA dataset does not have gender information. It has the sex of the donors. This should be corrected in both the manuscript and on the website.

We thank the reviewer for his/her positive views on the platform. We have changed “gender” to “sex” in the interface, associated datasets, and the manuscript.

Dear Dr. Ekiz,

Thank you for the submission of your revised manuscript. We have now received the enclosed reports from the referees that were asked to assess it and both support its publication now. Referee 2 still has a few more minor suggestions that I would like you to incorporate before we can proceed with the official acceptance of your manuscript.

A few editorial requests will also need to be addressed:

- Please separate the results and discussion sections. Your ms has 7 main figures and will be published as a full article.
- Please correct the conflict of interest subheading to "Disclosure and Competing Interests Statement"
- The author credits need to be removed from the ms file. All credits need to be entered during online ms submission.
- The reference format need to be changed to alphabetical, please use the EMBO reports reference style.
- Regarding FUNDING INFO: the text in the Comments box needs to be removed as the production only accepts separate entries; in addition, 2210-A graduate scholarship from TUBITAK (listed in the Comments box) needs to be provided among the separate entries; there is a discrepancy in 121C115 in the ms vs. 121C155 in our online submission system. Please correct.
- FIGURE CALLOUTS: "Supplement" and "Supplementary Information" should be removed or updated as we do not use this nomenclature.
- DATASET EV LEGENDS: there are 7 Tables uplidd separately as Excel files, but the nomenclature is not consistent: e.g. Table 1 vs. Table EV1; Table 1, 3, 5 and 6 should be called Datasets EV1-EV4 (ms callouts included); Tables 2, 4 and 7 should be renamed to Table EV1-EV3 (ms callouts included). Please correct all names in the files and in the ms text.
- APPENDIX FILE WITH ToC: in, but needs to be in PDF; the correct nomenclature throughout the file needs to have the word "Appendix" in all places: "Appendix Figure S1", etc.; the separately uplidd Appendix figures should be removed as we only need Appendix figures in one Appendix PDF file.
- Please upload a Reagents & Tools table with your final ms files. You can find a template for the table file in our guide to authors online under Reagents and Tools Table.
- Please upload the Source Data as 1 (zipped) folder per main figure.
- The manuscript sections should be in the following order: Title page - Abstract & Keywords - Introduction - Results - Discussion - Methods - Data Availability - Acknowledgments - Disclosure Statement & Competing Interests - References - Figure Legends - (Main Tables with legends if applicable) - Expanded View Figure Legends.
- Materials and Methods should be "Methods".

Figure Legends - Comments

- Please indicate what */ **/ ***/ **** represents; if this represents p value(s), please indicate the statistical test used and where appropriate and the exact p value in the legend(s) of figure(s) 7H
- Please note that exact p values (as reasonable) should be provided in the legends of figures 2C, 3A, B, D; 7G; EV1 B-D; EV2 D, E; EV4 A; EV5 C
- Please indicate the statistical test used for data analysis in the legends of figures 3D, E; 4E, F; 5D, E, I, J; EV2 D, E; EV5 B
- Please note that in figures 2C, 3A, 7G, EV5 C there is a mismatch between the annotated p values in the figure legend and the annotated p values in the figure file that should be corrected.
- Please note that the box plots need to be defined in terms of minima, maxima, centre, bounds of box and whiskers, and percentile in the legends of figures 2C, 3A, 6B, 7G, H; EV1 A-D; EV4 B, EV5 C.
- Please note that information related to n is missing in the legends of figures 2C, 3A, 5B, G; 6B, D; 7G, H; EV1 A-D; EV2 A, EV4 B, D; EV5 C.
- Please note that the error bars are not defined in the legends of figures 6D, 7B, EV4 D.

I would like to suggest some minor changes to the abstract that needs to be written in present tense:

Analyzing gene expression data from the Cancer Genome Atlas (TCGA) and similar repositories often requires advanced coding skills, creating a barrier for many researchers. To address this challenge, we developed The Cancer Genome Explorer (TCGEX),

a user-friendly, web-based platform for conducting sophisticated analyses such as survival modeling, gene set enrichment analysis, unsupervised clustering, and linear regression-based machine learning. TCGEx provides access to preprocessed TCGA data and immune checkpoint inhibition studies while allowing integration of user-uploaded datasets. Using TCGEx, we explore molecular subsets of human melanoma and identify microRNAs associated with intratumoral immunity. These findings are validated with independent clinical trial data on immune checkpoint inhibitors for melanoma and other cancers. Additionally, we identify cytokine genes predictive of treatment responses to various immune checkpoint inhibitors prior to treatment. Built on the R/Shiny framework, TCGEx offers customizable features to adapt analyses for diverse research contexts and generate publication-ready visualizations. TCGEx is freely available at <https://tcgex.iyte.edu.tr>, providing an accessible tool to extract insights from cancer transcriptomics data.

EMBO press papers are accompanied online by A) a short (1-2 sentences) summary of the findings and their significance, B) 2-3 bullet points highlighting key results and C) a synopsis image that is exactly 550 pixels wide and 200-600 pixels high (the height is variable). The synopsis image should provide a sketch of the major findings, like a graphical abstract. Please note that text needs to be readable at the final size. Please send us this information along with the final manuscript.

The synopsis image you sent is a little too crowded at the final image size and the font size is too small. It might be better to simplify the image. Please send us the new image at its final size of 550 pixels wide.

Referee #1:

I appreciate that the authors addressed most of the points I made in the comments. Especially, thank to the authors that they added the scRNA-seq analysis as a complementary to Figure 5. I don't have further major comments on the manuscript, and it would be great if this manuscript will be published in the current form.

Referee #2:

I thank authors for their thorough revisions. Most of my points were correctly addressed, with only minor corrections remaining.

Table 7 comparing TCGEX to competitor comes at the end of discussion. I think it should be presented at the beginning of results. Also, resource in the table are not clearly identified. What is "DB"? what is "Explorer"? "Portal" should be cBioPortal. References or links should be provided.

References (Carter et al, 2019; Auslander et al, 2019) are missing.

I did not catch the logic of supplementary figures classification, as either "Fig EVx" or "Appendix Fig Sx"

Dear Editor,

Manuscript is revised to address all of the editorial requests and the last comments from the reviewer-2. Please see a few comments below to specific requests (highlighted in blue). Summary sentences and bullet points are also included here.

Dear Dr. Ekiz,

Thank you for the submission of your revised manuscript. We have now received the enclosed reports from the referees that were asked to assess it and both support its publication now. Referee 2 still has a few more minor suggestions that I would like you to incorporate before we can proceed with the official acceptance of your manuscript.

A few editorial requests will also need to be addressed:

- Please separate the results and discussion sections. Your ms has 7 main figures and will be published as a full article.
- Please correct the conflict of interest subheading to "Disclosure and Competing Interests Statement"
- The author credits need to be removed from the ms file. All credits need to be entered during online ms submission.
- The reference format need to be changed to alphabetical, please use the EMBO reports reference style.
- Regarding FUNDING INFO: the text in the Comments box needs to be removed as the production only accepts separate entries; in addition, 2210-A graduate scholarship from TUBITAK (listed in the Comments box) needs to be provided among the separate entries; there is a discrepancy in 121C115 in the ms vs. 121C155 in our online submission system. Please correct. . Manuscript 121C115 is the correct grant number, and it is updated in the system
- FIGURE CALLOUTS: "Supplement" and "Supplementary Information" should be removed or updated as we do not use this nomenclature.
"Supplement" and "Supplementary Information" ->>> Appendix
- DATASET EV LEGENDS: there are 7 Tables uplidd separately as Excel files, but the nomenclature is not consistent: e.g. Table 1 vs. Table EV1; Table 1, 3, 5 and 6 should be called Datasets EV1-EV4 (ms callouts included); Tables 2, 4 and 7 should be renamed to Table EV1-EV3 (ms callouts included). Please correct all names in the files and in the ms text.
- APPENDIX FILE WITH ToC: in, but needs to be in PDF; the correct nomenclature throughout the file needs to have the word "Appendix" in all places: "Appendix Figure S1", etc.; the separately uplidd Appendix figures should be removed as we only need Appendix figures in one Appendix PDF file.
- Please upload a Reagents & Tools table with your final ms files. You can find a template for the table file in our guide to authors online under Reagents and Tools Table.
- Please upload the Source Data as 1 (zipped) folder per main figure. While we initially uploaded source data for some of the figures, our later correspondence indicated that source data isn't required since all the data used for generating the figures are easily accessible in the FigShare folder.
- The manuscript sections should be in the following order: Title page - Abstract & Keywords - Introduction - Results - Discussion - Methods - Data Availability - Acknowledgments - Disclosure Statement & Competing Interests - References - Figure Legends - (Main Tables with legends if applicable) - Expanded View Figure Legends.
- Materials and Methods should be "Methods".

Figure Legends - Comments

- Please indicate what */ **/ ***/ **** represents; if this represents p value(s), please indicate the statistical test used and where appropriate and the exact p value in the legend(s) of figure(s) 7H
- Please note that exact p values (as reasonable) should be provided in the legends of figures 2C, 3A, B, D; 7G; EV1 B-D; EV2 D, E; EV4 A; EV5 C
- Please indicate the statistical test used for data analysis in the legends of figures 3D, E; 4E, F; 5D, E, I, J; EV2 D, E; EV5 B
- Please note that in figures 2C, 3A, 7G, EV5 C there is a mismatch between the annotated p values in the figure legend and the annotated p values in the figure file that should be corrected.
- Please note that the box plots need to be defined in terms of minima, maxima, centre, bounds of box and whiskers, and percentile in the legends of figures 2C, 3A, 6B, 7G, H; EV1 A-D; EV4 B, EV5 C.
- Please note that information related to n is missing in the legends of figures 2C, 3A, 5B, G; 6B, D; 7G, H; EV1 A-D; EV2 A, EV4 B, D; EV5 C.
- Please note that the error bars are not defined in the legends of figures 6D, 7B, EV4 D.

I would like to suggest some minor changes to the abstract that needs to be written in present tense:

Analyzing gene expression data from the Cancer Genome Atlas (TCGA) and similar repositories often requires advanced coding skills, creating a barrier for many researchers. To address this challenge, we developed The Cancer Genome Explorer (TCGEx), a user-friendly, web-based platform for conducting sophisticated analyses such as survival modeling, gene set enrichment analysis, unsupervised clustering, and linear regression-based machine learning. TCGEx provides access to preprocessed TCGA data and immune checkpoint inhibition studies while allowing integration of user-uploaded datasets. Using TCGEx, we explore molecular subsets of human melanoma and identify microRNAs associated with intratumoral immunity. These findings are validated with independent clinical trial data on immune checkpoint inhibitors for melanoma and other cancers. Additionally, we identify cytokine genes predictive of treatment responses to various immune checkpoint inhibitors prior to treatment. Built on the R/Shiny framework, TCGEx offers customizable features to adapt analyses for diverse research contexts and generate publication-ready visualizations. TCGEx is freely available at <https://tcgex.iyte.edu.tr>, providing an accessible tool to extract insights from cancer transcriptomics data.

We thank the editor for this suggestion and we now revised the abstract as suggested.

EMBO press papers are accompanied online by A) a short (1-2 sentences) summary of the findings and their significance, B) 2-3 bullet points highlighting key results and C) a synopsis image that is exactly 550 pixels wide and 200-600 pixels high (the height is variable). The synopsis image should provide a sketch of the major findings, like a graphical abstract. Please note that text needs to be readable at the final size. Please send us this information along with the final manuscript.

The Cancer Genome Explorer (TCGEx) is a web-based visual interface that simplifies the analysis of cancer gene expression data from multiple sources by integrating sophisticated computational tools, including survival modeling, gene set enrichment analysis, and machine learning. Using TCGEx, we identified molecular markers associated with intratumoral immunity in

melanoma and validated them with independent clinical trial data, uncovering a cytokine signature predictive of positive immunotherapy responses.

1. TCGEx provides an intuitive, web-based platform that integrates statistical tools and machine learning for comprehensive analysis of cancer gene expression data without requiring coding expertise.
2. Our analyses revealed microRNAs associated with intratumoral immunity in TCGA melanoma and independent clinical trial data sets.
3. TCGEx unveiled a subset of cytokine genes predictive of positive responses to diverse immune checkpoint inhibitors prior to treatment initiation.

The synopsis image you sent is a little too crowded at the final image size and the font size is too small. It might be better to simplify the image. Please send us the new image at its final size of 550 pixels wide.

We revised the synopsis image and are providing it in the current submission as a png file. Side note on the figure fonts: We shrunk-down some of the figure panels from the paper just to demonstrate the kinds of graphs the user would see on the platform. We wanted to highlight the capabilities of the tool and included descriptive keywords (which should be large enough for readability) but the actual figure texts do not matter much for summarizing these capabilities. The biological findings are added as a summary at the bottom of the figure which should be still readable. We modeled our synopsis image after many others in the field and we believe it serves as a good summary for our manuscript.

Referee #1:

I appreciate that the authors addressed most of the points I made in the comments. Especially, thank to the authors that they added the scRNA-seq analysis as a complementary to Figure 5. I don't have further major comments on the manuscript, and it would be great if this manuscript will be published in the current form.

We appreciate the reviewer's positive assessment on the changes.

Referee #2:

I thank authors for their thorough revisions. Most of my points were correctly addressed, with only minor corrections remaining.

Table 7 comparing TCGEX to competitor comes at the end of diiscussion. I think it should be presented at the beginning of results. Also, ressource in the table are not clearly identified. What is "DB"? what is "Explorer"? "Portal" should be cBioPortal. References or links should be provided.

We thank the reviewer for finding our revisions adequate. We moved the table to the beginning of the results section as suggested. Additionally, we have added a row providing the accession links for each interface and clarified the names. We believe that the truncated names are an artifact of the PDF conversion in the system, and we provided an excel file to the EMBO Reports production for rendering the table legibly upon publication.

References (Carter et al, 2019; Auslander et al, 2019) are missing.

We apologize for the oversight. These references are now added.

I did not catch the logic of supplementary figures classification, as either "Fig EVx" or "Appendix Fig Sx"

To be perfectly frank, the convention of EMBO Reports for showing supplementary figures separately as "Extended View" and "Appendix" figures is somewhat new to us. After the initial submission where all supplementary figures were numbered traditionally, we wanted to benefit from this feature and selected some supplementary figures as EV (that we deemed more relevant to the main figures), and some others as Appendix figures (that provide additional data about the analyses). By rearranging supplementary figures this way, we hope to make supplementary data more readily accessible to the online readers.

Dr. Huseyin Ekiz
Izmir Institute of Technology
Department of Molecular Biology and Genetics
Gulbahce
Urla, Izmir
35430
Turkey

Dear Dr. Ekiz,

I am very pleased to accept your manuscript for publication in the next available issue of EMBO reports. Thank you for your contribution to our journal.
